# CDK4/6 inhibitors target SMARCA4-determined cyclin D1 deficiency in hypercalcemic small cell carcinoma of the ovary

Yibo Xue [ID] et al.[#]

Inactivating mutations in *SMARCA4* (*BRG1*), a key SWI/SNF chromatin remodelling gene, underlie small cell carcinoma of the ovary, hypercalcemic type (SCCOHT). To reveal its druggable vulnerabilities, we perform kinase-focused RNAi screens and uncover that SMARCA4-deficient SCCOHT cells are highly sensitive to the inhibition of cyclin-dependent kinase 4/6 (CDK4/6). SMARCA4 loss causes profound downregulation of cyclin D1, which limits CDK4/6 kinase activity in SCCOHT cells and leads to in vitro and in vivo susceptibility to CDK4/6 inhibitors. SCCOHT patient tumors are deficient in cyclin D1 yet retain the retinoblastoma-proficient/p16$^{INK4a}$-deficient profile associated with positive responses to CDK4/6 inhibitors. Thus, our findings indicate that CDK4/6 inhibitors, approved for a breast cancer subtype addicted to CDK4/6 activation, could be repurposed to treat SCCOHT. Moreover, our study suggests a novel paradigm whereby critically low oncogene levels, caused by loss of a driver tumor suppressor, may also be exploited therapeutically.

Cancer therapy is shifting towards genotype-based strategies, where signaling pathways activated by oncogenic mutations are targeted by highly selective inhibitors. However, a majority of cancers lack a known druggable driver oncogene or have lost tumor suppressors that are not directly actionable, thus remaining as a major clinical challenge.

Subunits of SWI/SNF chromatin remodeling complexes are mutated in >20% of human cancers, across a broad range of cancer types, highlighting their important roles in tumorigenesis[1–3]. SMARCA4, encoding a SWI/SNF catalytic ATPase subunit, is inactivated by mutations or other mechanisms in different cancers, including non-small cell lung cancer (NSCLC), breast cancer, glioblastoma, and others[4–7]. However, the underlying mechanisms of SMARCA4 loss in driving tumorigenesis are currently unclear. Thus SMARCA4-deficient cancers still lack rationalized and targeted treatment options.

We and others uncovered that small cell carcinoma of the ovary, hypercalcemic type (SCCOHT), a rare and often lethal cancer of young women, is almost always caused by biallelic deleterious mutations in SMARCA4, leading to loss of SMARCA4 protein expression[8–11]. SCCOHT is an aggressive cancer, with long-term survival rates at early stage diagnosis of 33% with current surgical and chemotherapy/radiation treatments[12]. In contrast to other ovarian cancer subtypes, SCCOHT has a remarkably simple genome that harbors few mutations or chromosomal alterations[13,14]. While SMARCA4 loss is not directly targetable, this monogenic nature of SCCOHT presents an ideal opportunity to uncover druggable targets that are synthetic lethal with SMARCA4 loss through functional genetic screens.

Here we employed this unbiased synthetic lethal approach and uncovered an unexpected molecular vulnerability of SMARCA4 deficiency in SCCOHT. We demonstrated that this vulnerability of SCCOHT can be effectively targeted by cyclin-dependent kinase 4/6 (CDK4/6) inhibitors, which have been approved by the U.S. Food and Drug Administration (FDA) for treating estrogen receptor-positive (ER+), human epidermal growth factor receptor 2-negative (HER2−) advanced breast cancers[15–19].

## Results

### SCCOHT cells are dependent on CDK4/6 kinase activities.
We set out to uncover vulnerabilities in SMARCA4-deficient SCCOHT using synthetic lethal screens, which are powerful tools to identify drug targets and help derive cancer-specific therapies that have minimal side effects in normal tissue[20,21]. Appropriate controls for the synthetic lethal screen would be isogenic SCCOHT cell lines engineered to express SMARCA4 or non-transformed cells of the same origin that are SMARCA4-proficient. In line with a previous report[22], we found that forced SMARCA4 expression in the two established SCCOHT cell lines, BIN-67[23] and SCCOHT-1[24], resulted in strong growth inhibition; this was not seen in the non-transformed ovarian epithelial IOSE80 cells that are SMARCA4-proficient (Supplementary Fig. 1). Similar SMARCA4-induced growth inhibition was also observed in COV434[25] (Supplementary Fig. 1), which was initially designated as an ovarian juvenile granulosa cell tumor line but recently redefined as a SCCOHT cell line[26] and has been verified in an independent study[27]. Therefore, isogenic controls may not be feasible for synthetic lethal screens in settings where SCCOHT cells require SMARCA4 loss for normal proliferation. Although the exact origin of SCCOHT is unknown, the tumors clearly arise from the ovary; thus IOSE80 was chosen as a SMARCA4-proficient screening control. OVCAR4, an ovarian carcinoma-derived cell line which is SMARCA4-proficient, was also included as an additional control.

Using our validated short hairpin RNA (shRNA) library targeting the human kinome[28–30], we performed pooled screens to identify kinases whose inhibition is selectively lethal to BIN-67 but not to IOSE80 and OVCAR4 (Fig. 1a). This focused library was chosen because pharmacological inhibitors targeting the kinases identified from our screens, if available, would have the highest chance of clinical implementation. In addition, incomplete gene suppression by RNA interference (RNAi) could provide a more realistic mimic of drug inhibition. Upon screen completion, we analyzed the data using the MAGeCK statistical software package[31] and identified CDK6 as the first ranked gene that was negatively selected in BIN-67 (Fig. 1b and Supplementary Data 1). In contrast, CDK6 was not significantly selected in IOSE80 and OVCAR4 control cells (Fig. 1b and Supplementary Data 1). Validating this, CDK6 knockdown caused a marked inhibition of proliferation in all three SCCOHT cell lines BIN-67, SCCOHT-1, and COV434 but did not significantly impact SMARCA4-proficient control cell lines IOSE80 and OVCAR4 (Fig. 1c, d). CDK6 and the closely related CDK4 are activated by forming complexes with D cyclins to phosphorylate and inhibit retinoblastoma (RB) protein, allowing cell cycle progression[16,18]. Consistent with this, CDK6 knockdown suppressed RB phosphorylation in SCCOHT cells but not in SMARCA4-proficient cells (Fig. 1d), supporting the decrease in proliferation observed.

From the same screen analysis, we noted that CDK4 was the second ranked lethal gene in BIN-67 and was also significantly selected in the control cells (Fig. 1b and Supplementary Data 1). In line with this, suppression of CDK4 expression using two independent shRNAs inhibited growth of all cell lines (Fig. 1c). However, RB phosphorylation was suppressed only in SCCOHT cells but not in SMARCA4-proficient controls upon CDK4 knockdown (Fig. 1d). These observations suggest that growth inhibition induced by CDK4 knockdown in SMARCA4-proficient controls is mediated by a kinase-independent activity of CDK4; in contrast, inhibition of CDK4/6 kinase activities in SCCOHT cells is likely to underlie the suppression of proliferation upon CDK4/6 knockdown.

Supporting this, reconstitution of wild-type CDK6 but not the kinase-inactive mutant CDK6D163N rescued the growth inhibition induced by CDK6 knockdown in SCCOHT cells (Fig. 1e, f). Similar results using wild-type CDK4 and the kinase-inactive mutant CDK4D158N were also obtained in SCCOHT cells (Fig. 1g, h). In contrast, both CDK4 constructs rescued growth inhibition induced by CDK4 knockdown in SMARCA4-proficient cells (Fig. 1i, j). Taken together, these findings indicate that SCCOHT cells are more vulnerable to inhibition of CDK4/6 kinase activities, compared to SMARCA4-proficient control cells.

### SCCOHT cells are highly sensitive to CDK6 inhibitors.
Three highly selective CDK4/6 inhibitors, palbociclib (PD-0332991), ribociclib (LEE001), and abemaciclib (LY2835219), have been recently approved by the FDA for treating ER+/HER2− advanced breast cancers, which are often characterized by dysregulated CDK4/6 activation[15–19]. In keeping with our above findings that SCCOHT cells are more susceptible to inhibition of CDK4/6 kinase activities compared to SMARCA4-proficient controls, we found that SCCOHT cells but not SMARCA4-proficient controls, including IOSE80, OVCAR4, and OVCAR8 (an additional ovarian carcinoma line), are highly sensitive to palbociclib in both colony-formation (Fig. 2a) and cell viability (Fig. 2b) assays. Furthermore, SCCOHT cells have similar or lower half maximal inhibitory concentration (IC50) compared to the control ER+ breast cancer cells MCF7 and CAMA-1 (Fig. 2a, b), the latter among the most palbociclib-sensitive lines in a panel of ~50 breast cancer cell lines[32]. Consistent with the growth response,

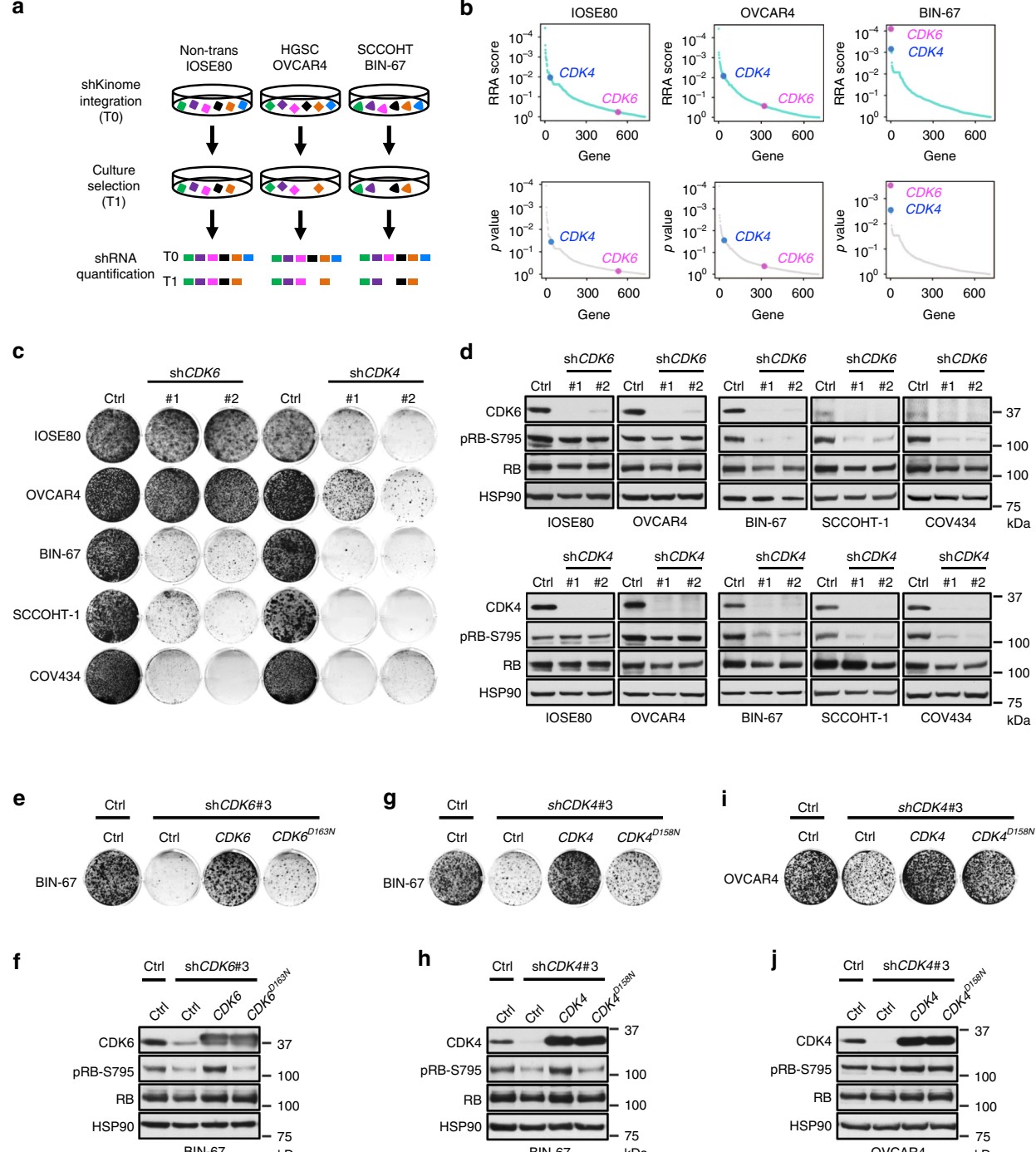

**Fig. 1** SMARCA4-deficient SCCOHT cells are vulnerable to inhibition of CDK4/6 kinase activities. **a** Schematic outline of the shRNA screens for kinases whose inhibition is selectively lethal to SMARCA4-deficient SCCOHT cells (BIN-67) but not to SMARCA4-proficient control cells (IOSE80, OVCAR4). Cells were infected with the lentiviral shRNA library (T0) and cultured for selection for 14 days (T1). The relative abundance of shRNAs in the cell populations was determined by next-generation sequencing. **b** Analysis of the shRNA screens using the MAGeCK statistical software package[31]. CDK6 (magenta) and CDK4 (blue) are the first two ranked genes that were negatively selected in BIN-67 cells. All genes were ranked based on their RRA (robust rank aggregation, top) or raw $p$ values (bottom) generated from the MAGeCK analysis. **c**, **d** Validation of CDK6 and CDK4 in SCCOHT cells (BIN-67, SCCOHT-1, COV434) and SMARCA4-proficient controls (IOSE80, OVCAR4). **c** Colony-formation assay of the indicated cell lines expressing pLKO control or shRNAs targeting CDK6 or CDK4 after 10–15 days of culturing. For each cell line, all dishes were fixed at the same time, stained, and photographed. **d** Western blot analysis of CDK6 and CDK4 and phosphorylated RB at serine 795 (pRB-S795) in the cells described in **c**. HSP90 was used as a loading control. **e–j** SCCOHT cells are more vulnerable to inhibition of CDK4/6 kinase activities, compared to SMARCA4-proficient control cells. **e** BIN-67 cells stably expressing pLX304-GFP, pLX304-CDK6, or pLX304-CDK6^D163N were infected with viruses containing pLKO control or a shRNA targeting the 3'UTR of CDK6, selected for integration, and cultured for 14 days. All dishes were fixed at the same time. **f** Western blot analysis for CDK6, pRB-S795, and HSP90 in the cells described above. **g**, **i** BIN-67 (**g**) and OVCAR-4 (**i**) cells expressing pLX317-GFP, pLX317-CDK4, or pLX317-CDK4^D158N were infected with viruses containing pLKO control or a shRNA vector targeting the 3'UTR of CDK4, selected for integration, and cultured for 14 days. For each cell line, all dishes were fixed at the same time. **h**, **j** Western blot analysis for CDK4, pRB-S795, and HSP90 in the cells described above

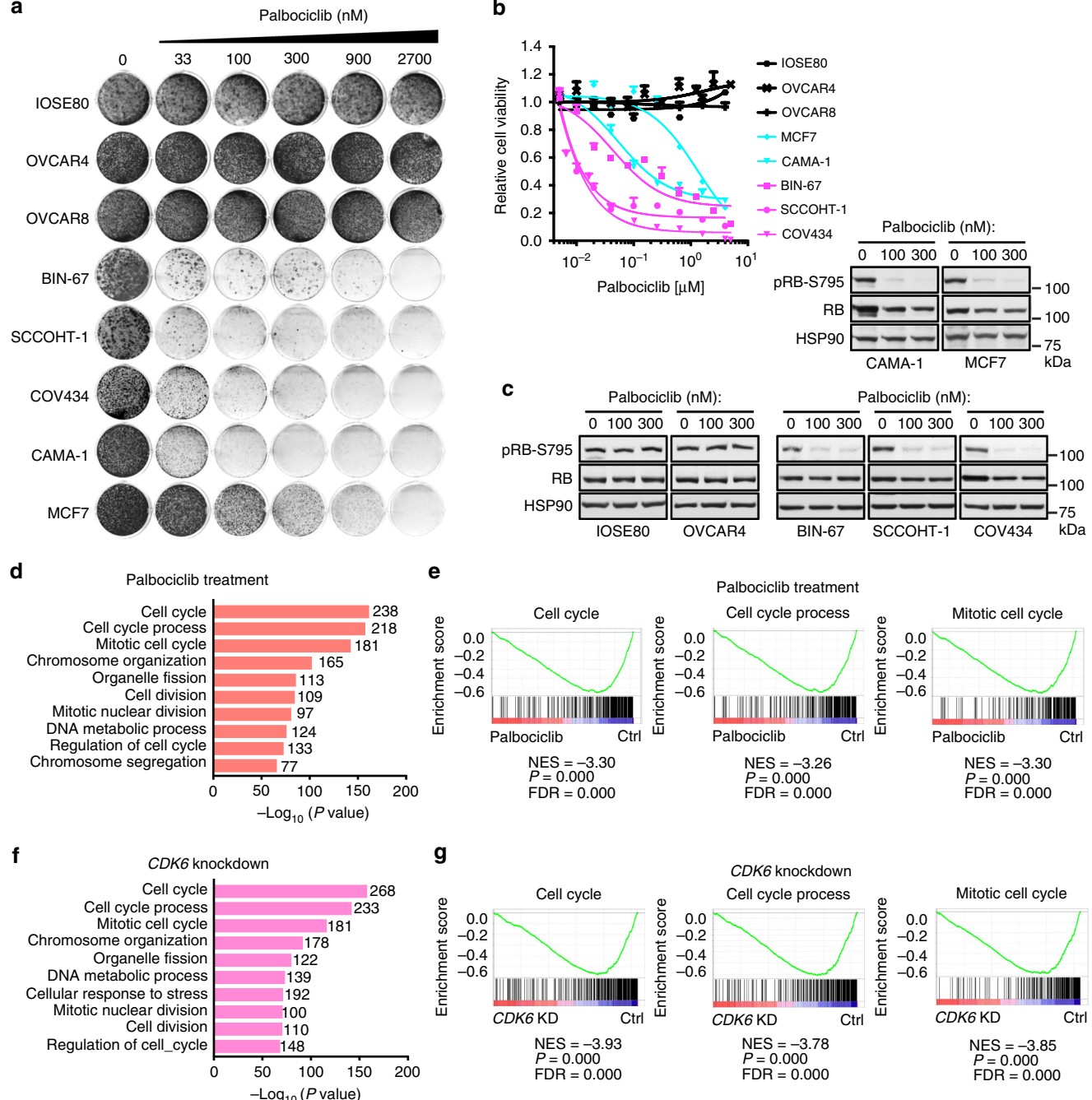

**Fig. 2** SCCOHT cells are highly sensitive to CDK4/6 inhibitors. **a–c** SMARCA4-deficient SCCOHT cells are highly sensitive to palbociclib treatment, similar to ER+ breast cancer cells. **a** Colony-formation assays in SCCOHT (BIN-67, SCCOHT-1, and COV434), SMARCA4-proficient ovarian (IOSE80, OVCAR4, and OVCAR8), and ER+ breast cancer (MCF7 and CAMA-1) cell lines. Cells were cultured in the absence or presence of palbociclib at the indicated concentrations for 10–21 days. For each cell line, all dishes were fixed at the same time. **b** Cell viability assay of the same cell line panel. Cells were treated with increasing concentrations of palbociclib for 5–10 days, and cell viability using CellTiter-Blue was determined by measuring the fluorescence (560/590 nm) in a microplate reader. Error bars: mean ± standard deviation (s.d.) of biological replicates ($n = 4$). **c** Palbociclib treatment suppresses RB phosphorylation in SCCOHT cells similar to ER+ breast cancer cells but not in IOSE80 and OVCAR4 cells. Levels of pRB-S795 in cells treated with 0, 100, and 300 nM of palbociclib for 24 h were documented by western blot analysis. **d**, **e** Transcriptome profiling in SCCOHT cells show that top ranked pathways affected upon palbociclib treatment are related to cell cycle regulation. RNA-Seq was performed in BIN-67 and SCCOHT-1 cells treated with 100 nM of palbociclib for 24 h. Common genes that significantly changed ($p < 0.05$) in both cell lines was analyzed using Gene Set Enrichment Analysis (GSEA). Top ten cellular processes by Gene Ontology (GO) term (**d**) and GSEA plots for the top three cellular processes (**e**) are shown. NES normalized enrichment score, FDR false discovery rate. **f**, **g** Transcriptome profiling in SCCOHT cells show that top ranked pathways affected upon CDK6 knockdown are related to cell cycle regulation. RNA-Seq was performed in BIN-67 and SCCOHT-1 cells expressing pLKO control or two independent shRNAs targeting CDK6. Common genes that significantly changed ($p < 0.05$) in both shRNAs and both cell lines were analyzed using GSEA. Top ten cellular processes by GO term (**f**) and GSEA plots for the top three cellular processes (**g**) are shown. NES normalized enrichment score, FDR false discovery rate

palbociclib suppressed RB phosphorylation in both SCCOHT and breast cancer cells but not in IOSE80 and OVCAR4 (Fig. 2c). Similar results were also obtained using abemaciclib and ribociclib (Supplementary Fig. 2). Next, we performed transcriptome analysis using RNA-Seq in BIN-67 and SCCOHT-1 cells treated with palbociclib or expressing shRNAs targeting *CDK6*. Gene set enrichment analysis (GSEA) show that the top ranked pathways affected in these cells upon palbociclib treatment or *CDK6* knockdown are all cell cycle process-related and closely mirror each other (Fig. 2d–g). Together, these data demonstrate that CDK4/6 inhibitors are highly effective in inhibiting proliferation of SCCOHT cell predominantly through cell cycle suppression.

**Cyclin D1 deficiency in SCCOHT drives the drug sensitivity.** To address the molecular mechanism underlying the susceptibility of SCCOHT cells to inhibition of CDK4/6 kinase activities, we examined the expression of key regulators of G1- to S-phase cell cycle progression in our cell line panel. In contrast to SMARCA4-proficient ovarian controls, SCCOHT and ER$^+$ breast cancer cell lines retain RB and express lower levels of the CDK4/6 inhibitor p16$^{INK4a}$ (Fig. 3a), a profile that has been associated with positive responses to palbociclib[15–19,33–35]. In addition, both SCCOHT and breast cancer cell lines express lower levels of CDK4 (Fig. 3a). Neither protein nor mRNA expression of CDK4/6 and other relevant cell cycle regulators such as cyclin E, p27, and p21 associate with SMARCA4 status (Fig. 3a, Supplementary Fig. 3a–f). Surprisingly, and in contrast with SMARCA4-proficient cells, all three SCCOHT cell lines express very low levels of cyclin D1 protein (Fig. 3a), which is associated with mRNA expressions of *CCND1* coding for cyclin D1 (Fig. 3b). We also examined other D cyclins in the cell line panel as they also complex with CDK4/6. SCCOHT-1 is the only cell line expressing cyclin D2 (Supplementary Fig. 3a, g). BIN-67 and SCCOHT-1, but not COV434, express lower levels of cyclin D3 compared to SMARCA4-proficient cells (Supplementary Fig. 3a, h). Therefore, cyclin D1 deficiency is the shared characteristic among the three SCCOHT cell lines and associates with SMARCA4 status.

We hypothesized that this cyclin D1 deficiency may limit the kinase activities of CDK4/6 in SCCOHT cells. To test this, we first performed immunoprecipitation to analyze the D cyclins associated with CDK4, whose expression is less variable among the cell lines compared to CDK6 (Fig. 3a). As expected, cyclin D1 was found to be in a complex with CDK4 in all SCCOHT cell lines and IOSE80 control (Supplementary Fig. 4a–d). Cyclin D3 was also found to complex with CDK4 in IOSE80 and COV434 cells but not in BIN-67 and SCCOHT-1 cells, likely due to its lower expression in the latter two cell lines (Supplementary Fig. 3a). Cyclin D2 was detected in the CDK4 immunoprecipitate of SCCOHT-1 (Supplementary Fig. 4c), suggesting that it may partially compensate for the low cyclin D1 expression in this cell line. However, our in vitro kinase assays, using the normalized amount of immunoprecipitated CDK4 complexes, show that phosphorylation of recombinant RB substrate was more efficient in IOSE80 cells compared to all SCCOHT cell lines, indicating lower total CDK4 kinase activity in SCCOHT cells (Fig. 3c). These data suggest that the common cyclin D1 deficiency of SCCOHT cells constrains their CDK4/6 activity, which could result in vulnerability to CDK4/6 inhibition.

Consistent with this, ectopic cyclin D1 expression in SCCOHT-1 and BIN-67 cells led to elevated RB phosphorylation (Fig. 3d, e; input lanes; Fig. 3f, g) associated with a proportional increase of cyclin D1–CDK4 complex (Fig. 3d, e; IP lanes), indicating increased CDK4 kinase activities. Furthermore, ectopic expression of cyclin D1 but not CDK4, substantially increased RB phosphorylation and conferred substantial resistance to CDK4/6

inhibitors in SCCOHT-1 and BIN-67 cells (Fig. 3f–i and Supplementary Fig. 5). Complementarily, we found that cyclin D1 and RB phosphorylation were elevated in two spontaneous drug-resistant clones, SCCOHT-1 R1 and R2, selected by prolonged palbociclib exposure (Fig. 3j, k). Moreover, cyclin D1 knockdown in SCCOHT-1 R1 cells reduced RB phosphorylation and re-sensitized these cells to palbociclib (Fig. 3l, m). Together, these data establish that cyclin D1 deficiency drives the vulnerability of SCCOHT cells to CDK4/6 inhibition.

**Cyclin D1 deficiency in SCCOHT is induced by SMARCA4 loss.** In parallel to the above described biochemical analysis, we performed RNA-Seq transcriptome profiling in BIN-67 and SCCOHT-1 cells before and after SMARCA4 restoration to unbiasedly uncover the genes and pathways dysregulated owing to SMARCA4 loss. Consistent with the previous established role of SMARCA4 as transcriptional activator[36], we found that SMARCA4 restoration predominantly upregulates gene expression—while 662 common genes are upregulated (Fig. 4a, Supplementary Data 2), only 5 common genes are downregulated in both cell lines (Fig. 4b, Supplementary Data 2). GSEA show that the top ranked pathways affected by SMARCA4 restoration are not directly related to cell cycle regulation (Supplementary Fig. 6). However, we identified *CCND1* as the only direct cell cycle regulator among the 300 top ranked genes upregulated upon SMARCA4 restoration (#21; Fig. 4c and Supplementary Data 2).

SMARCA4 has been linked to cyclin D1 regulation in other cancer types with opposing effects: SMARCA4 suppresses *CCND1* in MCF7 ER$^+$ breast cancer cells[37], whereas *SMARCA4* knockdown leads to downregulation of cyclin D1 protein in triple-negative breast cancer and glioma cells[38,39]. We found that SMARCA4 restoration in all three SCCOHT cell lines strongly elevated cyclin D1 mRNA and protein expression (Fig. 4d, e). In contrast, SMARCA4 restoration did not alter CDK6 expression in SCCOHT cells (Supplementary Fig. 7a). Conversely, *SMARCA4* knockdown in IOSE80 and OVCAR4 cells resulted in a strong downregulation of cyclin D1 mRNA and protein (Fig. 4f, g), further supporting the role of SMARCA4 in activating cyclin D1 expression. As anticipated from our RNA-Seq analysis (Fig. 4c and Supplementary Data 2), this regulation of cyclin D1 by SMARCA4 in these cell line pairs was not observed for other relevant cell cycle genes including *CCND3*, *CCNE1*, *CDKN1B*, and *CDKN1A* (Supplementary Fig. 8), although *CCNE1* and *CDKN1A* have been shown to be regulated by SMARCA4 in other cancer types, likely due to context dependency[40,41].

The substantial elevation of *CCND1* mRNA and the chromatin remodeling role of SMARCA4 suggest that SMARCA4 can directly regulate *CCND1* transcription. Indeed, chromatin immunoprecipitations (ChIPs) in SMARCA4-restored SCCOHT-1 and BIN-67 cells showed a significant SMARCA4 occupancy at the *CCND1* promoter but not at the control upstream regions of *CCND1* locus or promoters of *CCND3* and *CCNE1* (Fig. 4h). Conversely, SMARCA4 occupancy at *CCND1* promoter, but not at the same control regions, was significantly reduced in IOSE80 and OVCAR4 cells upon SMARCA4 knockdown (Fig. 4i). Supporting our findings, publicly available SMARCA4 ChIP-Seq data sets of eight cell lines of different tissue origins also show consistent SMARCA4 occupancy at the *CCND1* promoter region[42–49] (Supplementary Fig. 9). Together, these results are consistent with the model that SMARCA4 is an activator of *CCND1* transcription in the ovarian context.

To further support the role of SMARCA4 in cyclin D1 regulation and drug response to CDK4/6 inhibitor, we introduced low levels of exogenous SMARCA4 expression using a doxycycline-controlled expression system (Supplementary

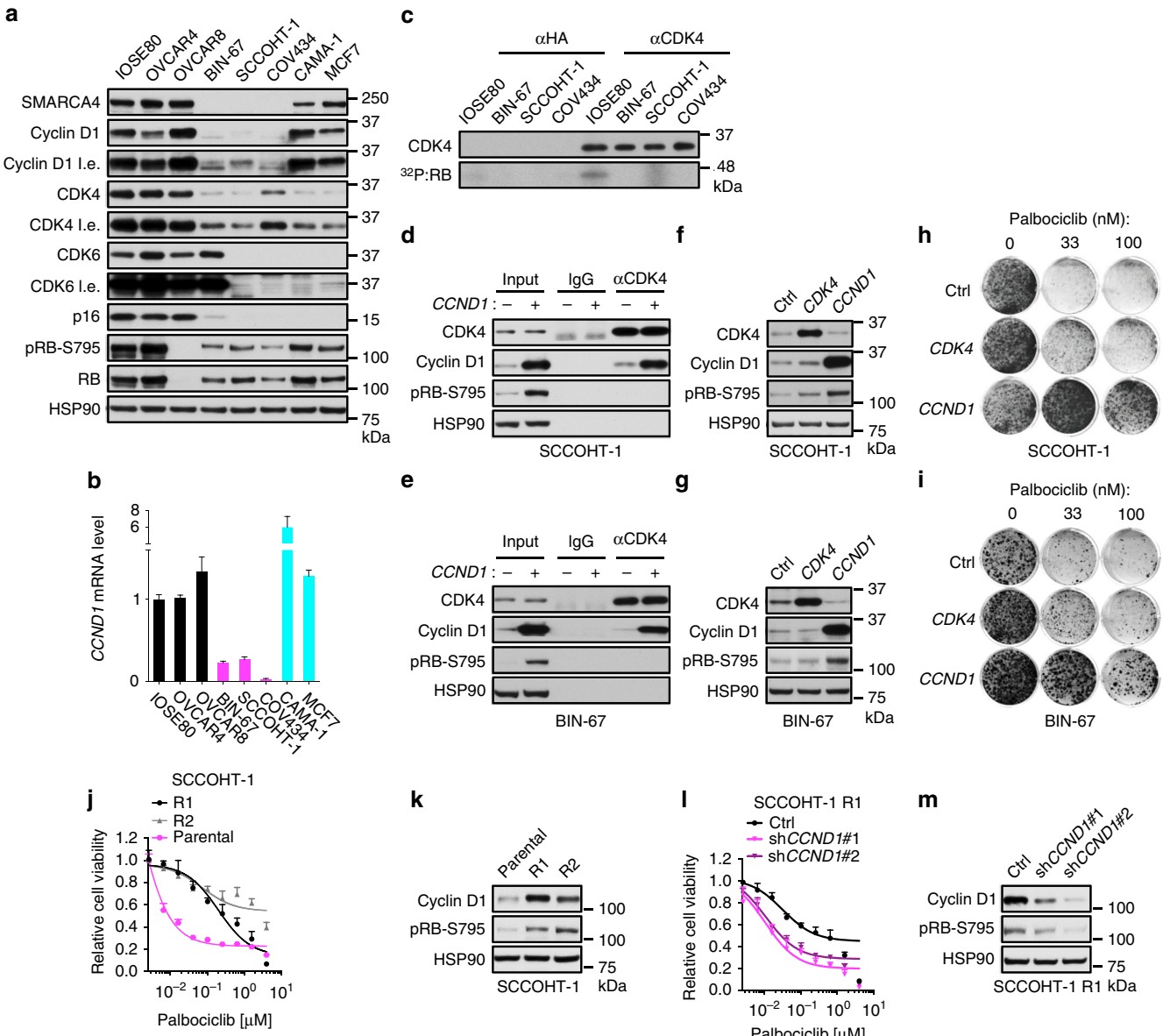

**Fig. 3** Cyclin D1 deficiency in SCCOHT cells results in vulnerability to CDK4/6 inhibition. **a** SMARCA4 loss in SCCOHT is associated with cyclin D1 deficiency and reduced CDK4 expression. Western blot analysis of key cell cycle regulators in a cell line panel: non-transformed ovarian epithelial (IOSE80), ovarian carcinoma (OVCAR4, OVCAR8), SCCOHT (BIN-67, SCCOHT-1, COV434), and ER+ breast cancer (CAMA-1, MCF7). l.e. long exposure. **b** SCCOHT cells express low levels of *CCND1* mRNA. Relative *CCND1* expression (normalized to *GAPDH*) in the cell panel described above were measured by qRT-PCR. Error bars: mean ± s.d. of biological replicates (*n* = 3). **c** SCCOHT cells have lower total CDK4 kinase activities compared to SMARCA4-proficient control cells. Normalized amount of immunoprecipitated CDK4 kinase complexes from IOSE80 and SCCOHT cell lines were subjected to in vitro kinase assays using recombinant RB. Upper, western blot analysis of immunoprecipitated CDK4 input; lower, kinase assay radiography. **d**–**i** Ectopic cyclin D1 expression increases RB phosphorylation and confers resistance to palbociclib in SCCOHT cells. **d**, **e** Western blot analysis of immunoprecipitations using an antibody against CDK4 or IgG in SCCOHT-1 (**d**) and BIN-67 (**e**) cells stably expressing *GFP* or *CCND1*. **f**, **g** Western blot analysis of SCCOHT-1 (**f**) and BIN-67 (**g**) cells with stable ectopic expression of *GFP*, *CDK4*, or *CCND1*. **h**, **i** Colony-formation assay of SCCOHT-1 (**h**) and BIN-67 (**i**) cells (described in **f**, **g**) treated with palbociclib. **j**, **k** Spontaneously palbociclib-resistant clones of SCCOHT-1 expressed elevated cyclin D1 and RB phosphorylation. **j** Cell viability assay of SCCOHT-1 parental cells and resistant clones (R1 and R2) treated with palbociclib for 9 days. Error bars: mean ± standard deviation (s.d.) of biological replicates (*n* = 4). **k** Western blot analysis for the indicated proteins in the cells described above. **l**, **m** Cyclin D1 knockdown in palbociclib-resistant SCCOHT-1 cells resensitizes them to palbociclib. **l** Cell viability assay of SCCOHT-1 R1 cells expressing pLKO control or two independent shRNAs targeting cyclin D1 treated with palbociclib for 9 days. Error bars: mean ± standard deviation (s.d.) of biological replicates (*n* = 4). **m** Western blot analysis for the indicated proteins in the cells described above

Fig. 10). We found that minimum levels of leaky SMARCA4 expression in BIN-67 and SCCOHT-1 cells (compared to IOSE80 control) were sufficient to upregulate cyclin D1 expression (Supplementary Fig. 10a, b, e, f), demonstrating the robust regulation by SMARCA4. Even though full expression of

SMARCA4 in SCCOHT cells strongly inhibits growth in long-term cultures (Supplementary Fig. 1), such low levels of SMARCA4 restoration in BIN-67 and SCCOHT-1 cells was tolerable (Supplementary Fig. 10c, g) and partially rescued these cells from the growth inhibition of palbociclib in a short-term

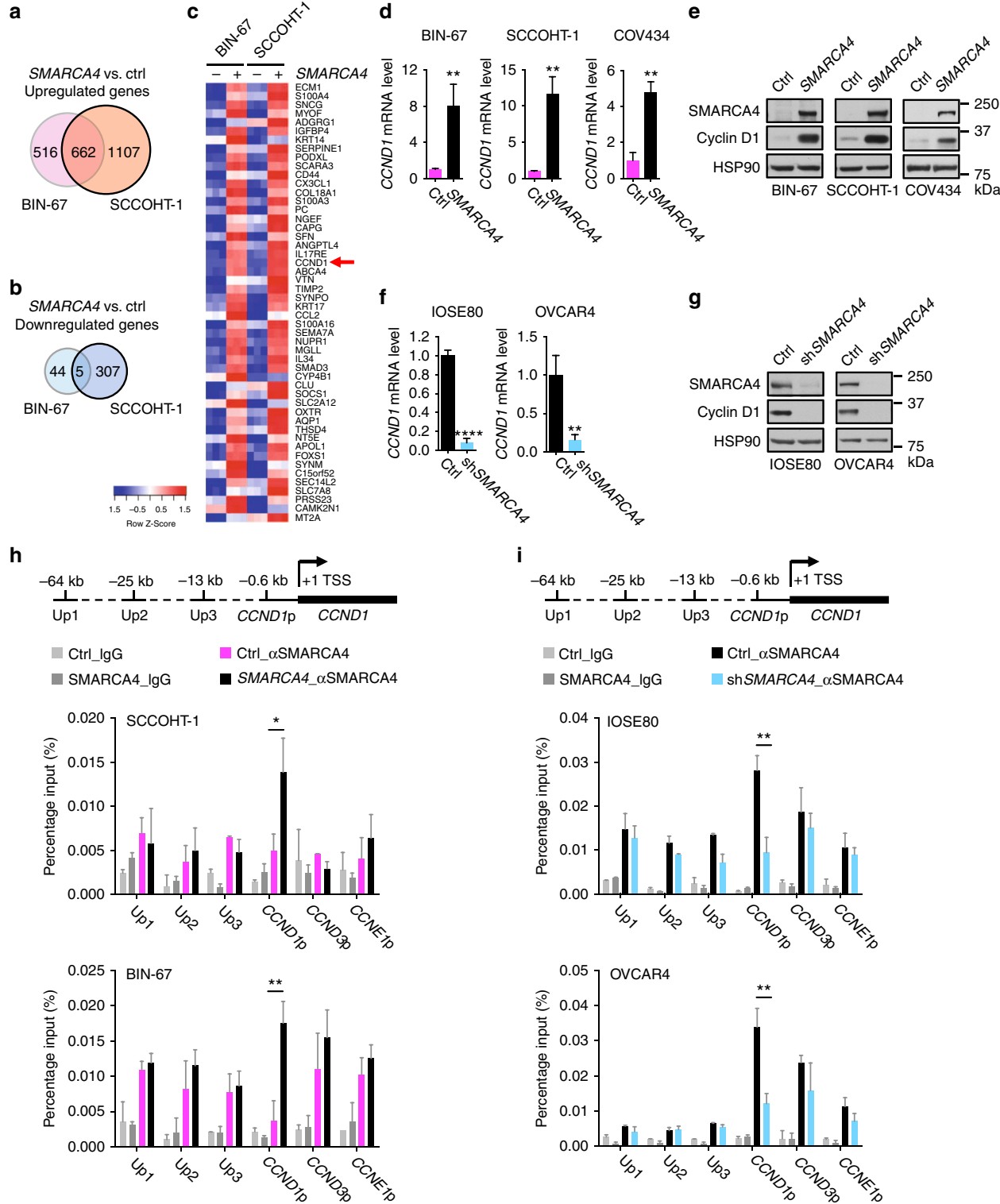

growth assay (Supplementary Fig. 10d, h). These data support the model that SMARCA4 loss in SCCOHT cells results in profound downregulation of cyclin D1 and is synthetic lethal with CDK4/6 inhibition.

**Palbociclib is effective against SCCOHT tumor growth in vivo.** Our data suggest that SMARCA4-dependent cyclin D1 deficiency constrains CDK4/6 activity and leads to the selective sensitivity to the CDK4/6 inhibition in SCCOHT cells. To establish this

potential treatment strategy in vivo, we examined the responses of SCCOHT tumors to the FDA-approved palbociclib in mouse xenograft settings. As shown in Fig. 5a, b, palbociclib treatment elicited a potent growth inhibition of BIN-67 tumors. Similar findings were obtained in a second xenograft model of SCCOHT-1 cells during the treatment course of 42 days (Fig. 5c, d). Furthermore, the immunohistochemical (IHC) analysis of tumors isolated at the treatment end points showed that RB phosphorylation, Ki67 expression, and mitotic index were significantly suppressed in palbociclib-treated cohorts (Fig. 5e–h), confirming

**Fig. 4** Cyclin D1 deficiency in SCCOHT cells is caused by SMARCA4 loss. **a–c** RNA-Seq analysis in BIN-67 and SCCOHT-1 cells stably expressing pReceiver control or pReceiver-*SMARCA4* identified *CCND1* as the top ranked cell cycle regulator upregulated upon SMARCA4 restoration (*n* = 3). **a, b** Venn diagrams showing the genes upregulated (**a**) or downregulated (**b**) upon SMARCA4 restoration (fold change >3, adjusted *p* < 0.05). **c** Heatmap of the top 50 genes upregulated upon SMARCA4 restoration in both BIN-67 and SCCOHT-1 cells. Red arrow points to *CCND1*. **d, e** SMARCA4 restoration in SCCOHT cells upregulated *CCND1* mRNA (**d**) and cyclin D1 protein (**e**) levels. **d** Relative expression levels of *CCND1* mRNA (normalized to *GAPDH*) in SCCOHT cells were measured by qRT-PCR. Error bars: mean ± s.d. of biological replicates (*n* = 3; two-tailed *t* test, **\*\****p* < 0.01). **e** Western blot analysis for the indicated proteins in the cells described above. **f, g** SMARCA4 knockdown in SMARCA4-proficient cells suppressed *CCND1* mRNA (**f**) and cyclin D1 protein (**g**) levels. **f** Relative expression levels of *CCND1* mRNA (normalized to *GAPDH*) in IOSE80 and OVCAR4 cells were measured by qRT-PCR. Error bars: mean ± s.d. of biological replicates (*n* = 3; two-tailed *t* test, **\*\*\*\****p* < 0.0001, **\*\****p* < 0.01). **g** Western blot analysis for the indicated proteins in the cells described above. **h, i** SMARCA4 occupancy in the *CCND1* promoter region. Chromatin immunoprecipitation experiments were performed in SCCOHT cells (SCCOHT-1, BIN-67) expressing pReceiver or pReceiver-*SMARCA4* (**h**) and in SMARCA4-proficient cells (IOSE80, OVCAR4) expressing pLKO or shRNA targeting *SMARCA4* (**i**), using an antibody against SMARCA4 or IgG. qPCR was used to analyze SMARCA4 occupancy using the primer sets for *CCND1*, *CCND3*, and *CCNE1* locus as indicated. p promoter, TSS transcription start site, error bars: mean ± s.d. of measurement replicates of a representative experiment (*n* = 3; two-tailed *t* test, **\****p* < 0.05, **\*\****p* < 0.01)

the target modulation by palbociclib. Moreover, we further evaluated the potential efficacy of palbociclib using a patient-derived xenograft (PDX) model of SCCOHT. As shown in Fig. 5i, palbociclib treatment also significantly suppressed the growth of SCCOHT PDXs. Together, these results establish that palbociclib is effective in treating SCCOHT tumors in vivo.

**SCCOHT patient tumors are deficient in cyclin D1 expression**. The reduced cyclin D1 expression in SCCOHT cells result in their sensitivity to CDK4/6 inhibitors (Fig. 3, Supplementary Fig. 5). Therefore, we analyzed the expression of cyclin D1 and other relevant cell cycle regulators in multiple patient tumor collections to evaluate the potential clinical implications of our findings. We first examined *CCND1* mRNA expression obtained from a NanoString gene expression study in 17 SCCOHTs[8,50] and 6 ovarian high-grade serous carcinomas (HGSCs). Consistent with the above-described cell line data (Fig. 3b), SCCOHT tumor samples expressed significantly lower *CCND1* levels compared to HGSCs (Fig. 6a, b). One of the three SCCOHT cell lines expressed *CCND2* (Supplementary Fig. 3a, g). In line with this, SCCOHT tumor samples expressed variable levels of *CCND2*, in contrast to HGSCs, which all exhibited low *CCND2* expression (Supplementary Fig. 10a, b). The NanoString data set also shows that these SCCOHT samples expressed higher *CDK4* levels than HGSCs (Supplementary Fig. 11a, b). While elevated *CDK4* mRNA expression was observed in two of the three SCCOHT cell lines (~1.5-fold compared to controls, Supplementary Fig. 3b), CDK4 regulation by SMARCA4 requires further investigations as SMARCA4 perturbation in SCCOHT and ovarian control cell lines yielded inconsistent results (Supplementary Fig. 7a–d). To validate the above-mentioned gene expression study, we performed quantitative PCR (qPCR) using available fresh frozen patient material from an independent series of HGSCs (*n* = 7) and SCCOHTs (*n* = 5). These SCCOHTs also expressed significantly lower levels of *CCND1*, but not *CCND2* and *CDK4*, compared to HGSCs (Fig. 6c and Supplementary Fig. 11c). Given that *CCND1* is an ER target gene[51], we evaluated the potential contribution of ER to the differential *CCND1* expression observed. We found that four of the seven HGSCs expressed similar levels of *ESR1* as SCCOHTs (Supplementary Fig. 10d) and that *CCND1* but not *CDK4* expression in these HGSCs was still significantly higher than SCCOHTs (Supplementary Fig. 10e), suggesting that *ESR1* is not a confounder of our analysis. Together, these tumor gene expression data support that SMARCA4 loss results in reduced *CCND1* expression in SCCOHT.

Using IHC coupled with unbiased automated quantification[52], we next analyzed the protein expression of these key cell cycle regulators in patient tumor samples of SCCOHT or HGSC embedded in tissue microarrays (TMAs). Figure 6d, e show that

SCCOHT tumors (*n* = 32) expressed significantly lower levels of cyclin D1 compared to HGSCs controls (*n* = 52) (~14-fold median difference, Supplementary Data 3). Similarly, SCCOHTs also expressed significantly lower levels of CDK4 compared to HGSCs with a ~29-fold median difference (Supplementary Fig. 11f, g and Supplementary Data 3). SCCOHT tumors showed heterogeneous CDK6 expression, similar to HGSCs (Supplementary Fig. 11f, g), which is in line with our previous observation that SMARCA4 restoration did not alter CDK6 expression in SCCOHT cells (Supplementary Fig. 7a). Additional IHC analysis also showed that SCCOHTs were generally RB-proficient and p16-deficient, which is the reverse of what is observed in HGSCs (Fig. 6d, e). These patient tumor IHC data are in support of our in vitro findings and confirm that cyclin D1 deficiency and lower CDK4 expression is a unique feature of SCCOHT; these samples also retained the known RB/p16 profile associated with positive responses to palbociclib[15–19,33–35]. Collectively, these results support the notion that CDK4/6 inhibitors could be effective in treating SCCOHT patients.

**Discussion**
Our study highlights the power of functional genetic screens in revealing unexpected cancer vulnerabilities to devise potential effective treatment strategies, especially in the context of tumors with quiescent genomes. Using this unbiased approach, we uncovered that cyclin D1 deficiency in SCCOHT results in exquisite susceptibility to CDK4/6 inhibitors.

First, our kinome-focused RNAi screens identified that SCCOHT cells are selectively sensitive to CDK4/6 knockdown. Subsequent rescue experiments using wild-type and kinase-inactive mutants of CDK4/6 show that SCCOHT cells are more dependent on CDK4/6 kinase activity than are SMARCA4-proficient controls. In line with this, SCCOHT cells are highly sensitive to CDK6 inhibitors both in vitro and in vivo. Mechanistically, SMARCA4 loss causes cyclin D1 deficiency, which limits CDK4/6 kinase activity in SCCOHT cells and results in less buffering against CDK4/6 inhibition. This is supported by multiple lines of evidence: immunoprecipitations and in vitro kinase assays indicate that cyclin D1 deficiency of SCCOHT cells constrains their CDK4/6 activity; ectopic expression of cyclin D1 in SCCOHT cells led to elevated RB phosphorylation and resistance to CDK4/6 inhibition; spontaneous palbociclib-resistant clones of SCCOHT cells expressed elevated cyclin D1 and RB phosphorylation, while cyclin D1 knockdown re-sensitized these cells to palbociclib. Finally, we show that SCCOHT patient tumors, similar to cell lines, also expressed reduced *CCND1* mRNA and cyclin D1 proteins yet retain the RB-proficient/p16-deficient profile, indicating that SCCOHT patients may benefit from CDK4/6 inhibitor treatment.

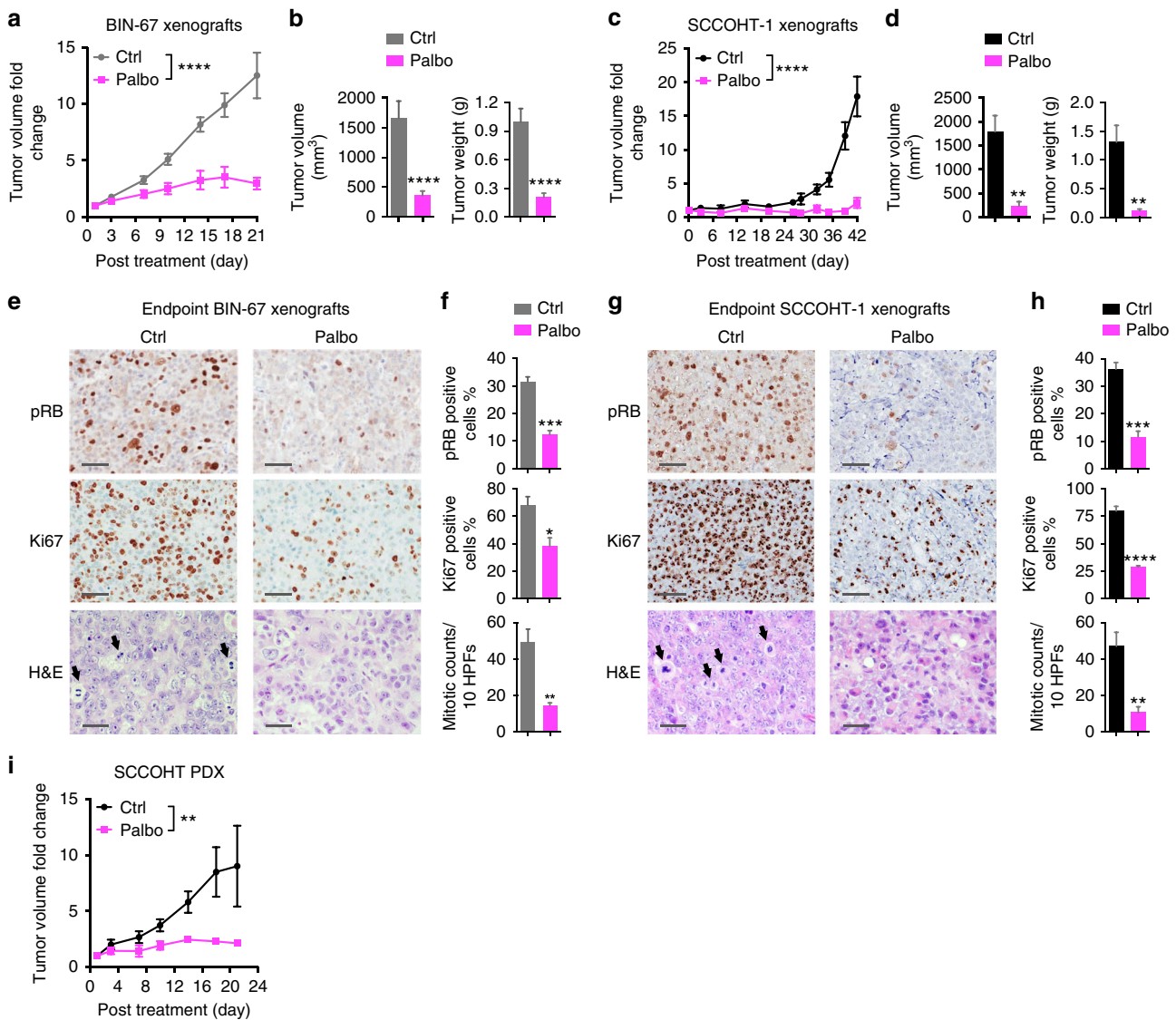

**Fig. 5** Palbociclib is effective in suppressing SCCOHT tumor growth in vivo. Palbociclib significantly suppresses tumor growth in xenograft models of BIN-67 (**a**, **b**, **e**, **f**) and SCCOHT-1 (**c**, **d**, **g**, **h**), as well as a patient-derived xenograft (PDX) model of SCCOHT (**i**). After tumor establishment (3–4 weeks for BIN-67 and SCCOHT-1; 3 months for the PDX), mice were treated daily with vehicle (ctrl) or palbociclib (palbo) for the indicated days. **a**, **c** Tumor volume fold change from Day 1 of treatment in BIN-67 (**a**, $n = 5$ per group) and SCCOHT-1 (**d**, $n = 4$ for vehicle, $n = 5$ for palbociclib; 150 mg kg$^{-1}$) models (two-way ANOVA, ****$p < 0.0001$). **b**, **d** The final tumor volume and weight measured at necropsy in BIN-67 (**b**, $n = 8$ per group) and SCCOHT-1 (**d**, $n = 4$ for vehicle, $n = 5$ for palbociclib) models (two-tailed $t$ test, ****$p < 0.0001$, **$p < 0.01$). **e**–**h** Palbociclib treatment resulted in suppression of RB phosphorylation, Ki67 expression, and mitotic index in BIN-67 and SCCOHT-1 xenograft tumors of the trial end points. Representative images of IHC (p-RB, Ki67) and hematoxylin and eosin (H&E) analysis of BIN-67 (**e**) and SCCOHT-1 (**g**) xenograft tumor tissues. Quantification results of p-RB, Ki67, and mitotic count of BIN-67 (**f**, $n = 3$) and SCCOHT-1 (**h**, $n = 4$). In the H&E images, black arrows point to mitotic active cells as an example. Bar 50 µm; two-tailed $t$ test, *$p < 0.05$, **$p < 0.01$, ***$p < 0.001$, ****$p < 0.0001$. **i** Tumor volume fold change from Day 1 of treatment in the SCCOHT PDX model ($n = 6$ for vehicle, $n = 3$ for palbociclib). Mice were treated with the initial dose of 150 mg kg$^{-1}$ of palbociclib. After first signs of any animal discomfort, we reduced the dose to 100 mg kg$^{-1}$. Mice that showed weight loss were not included for the analysis. Two-way ANOVA, **$p < 0.01$. Error bars: mean ± standard error of mean (s.e.m.)

It is possible that cyclin D1 deficiency in SCCOHT may be compensated by other regulators of cell cycle progression. For example, we observed elevated cyclin D2 expression in one of the three SCCOHT cell line (SCCOHT-1) as well as in some SCCOHT tumors. However, the in vitro kinase assays indicate that SCCOHT-1 cells, despite of elevated cyclin D2 expression, still have lower total CDK4 kinase activity compared to SMARCA4-proficient cells. This suggests that cyclin D2 elevation cannot fully compensate for cyclin D1 deficiency. Although we have not identified alterations in other key cell cycle regulators, our data do not rule out other potential dysregulations of cell

cycle progression in SCCOHT. In addition to cyclin D1, other factors associated with SMARCA4 loss may also contribute to drug sensitivities of SCCOHT cells to CDK4/6 inhibition. Furthermore, activation of other oncogenic pathways caused by SMARCA4 loss can drive malignant transformation[2,3], which remains to be investigated in SCCOHT.

Our unexpected findings contrast with the initial application of CDK4/6 inhibitors in treating ER$^+$ breast cancers that are often characterized with dysregulated CDK4/6 activation[15–19,33–35], where the oncogenic addiction to cyclin D1 is being targeted. However, the role of cyclin D1 in modulating response to CDK4/

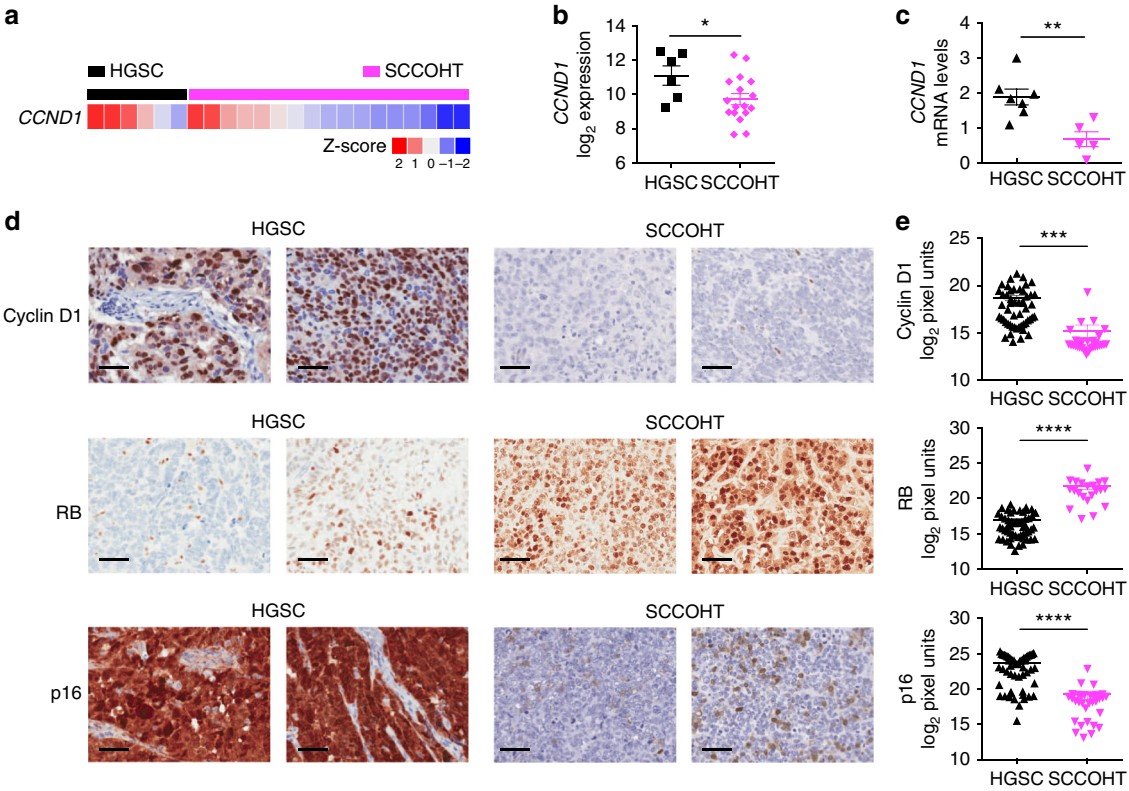

**Fig. 6** SCCOHT patient tumors expressed reduced levels of cyclin D1 mRNA and protein. **a**, **b** SCCOHT tumors expressed significantly lower *CCND1* mRNA levels compared to ovarian high-grade serous carcinomas (HGSCs). Heatmaps (**a**) and boxplots (**b**) showing *CCND1* and *CDK4* mRNA levels obtained from a NanoString gene expression study in SCCOHT patient tumors ($n = 17$) relative to HGSCs ($n = 6$). Two-tailed $t$ test, *$p < 0.05$. **c** qRT–PCR analysis of an independent cohort of fresh-frozen patient tumor samples show that SCCOHT ($n = 5$) expressed significantly low levels of *CCND1* mRNA (normalized to *GAPDH*) compared to HGSCs ($n = 7$). Two-tailed $t$ test, **$p < 0.01$; ns not significant. **d**, **e** SCCOHT patient tumors express low levels of cyclin D1 and retain the RB-proficient/p16-deficient profile associated with positive responses to palbociclib. Immunohistochemistry (IHC) analysis coupled with unbiased automated quantification[52] were performed on formalin-fixed paraffin-embedded HGSC ($n = 52$) and SCCOHT ($n = 32$; 4 of which were also analyzed by qRT–PCR in Fig. 6c) patient tumors for the expression of indicated key regulators of G1- to S-phase cell cycle. Representative images of the IHC analysis (**d**) and quantification results (**e**) are shown. Bar 50 μm; non-parametric Mann–Whitney test, ***$p < 0.001$, ****$p < 0.0001$; Error bars: mean ± standard error of mean (s.e.m.)

6 inhibitors is more complex than previously appreciated. A recent in vitro study across >500 cell lines suggests that cancer cells with activating *CCND1* genetic alternations (e.g., amplification, translocation) are sensitive to abemaciclib; paradoxically, *CCND1* mRNA is positively correlated with IC$_{50}$ of abemaciclib and high cyclin D1 protein expression is also weakly associated with abemaciclib resistance[53]. The latter finding of this study is consistent with our data, which provides an explanation for low expression levels of cyclin D1 in driving drug sensitivity to CDK4/6 inhibitors in certain contexts. Independently corroborating our findings in SCCOHT, we demonstrate that SMARCA4 loss in NSCLC also results in cyclin D1 deficiency and is synthetic lethal with CDK4/6 inhibition[54].

Our data demonstrate that cyclin D1 deficiency is a druggable vulnerability of SCCOHT, providing a rationale for a clinical trial using the available CDK4/6 inhibitors to treat this often-fatal cancer of young women. SCCOHT patients may also benefit from the antitumor immunity triggered by CDK4/6 inhibition as recently shown by others[55,56]. Finally, our study reveals a novel paradigm whereby a critically low level of an oncogene such as cyclin D1, caused by loss of a driver tumor suppressor, may also be a cancer vulnerability that can be targeted therapeutically.

## Methods

**Cell culture and viral transduction.** The BIN-67 cell line was obtained from Dr. S. R. Goldring (Hospital for Special Surgery, New York). OVCAR4 was from Dr. E.

Wang (University of Calgary, Calgary). BIN-67, SCCOHT-1[24], OVCAR4, and OVCAR8 were all cultured in RPMI with 7% fetal bovine serum (FBS), 1% penicillin/streptomycin, and 2 mM L-glutamine. IOSE80 cell line was obtained from Dr. Nelly Auersperg and was cultured in Medium 199/ MCDB 105 Medium (Sigma) with 5% FBS. COV434 was purchased from Sigma; OVCAR8, MCF7, and CAMA-1 cells were from ATCC and were chosen as representative breast cancer cells to direct compare the drug sensitivities with other cell lines used in our study; these cell lines were cultured in Dulbecco's modified Eagle's medium with 7% FBS, 1% penicillin/streptomycin, and 2 mM L-glutamine. All cell lines were free of mycoplasma and were maintained at 37 °C and 5% CO$_2$. All cell lines have been validated by Short-Tandem Repeat profiling and regularly tested for mycoplasma using the Mycoalert Detection Kit (Lonza).

Lentiviral transduction was performed using the protocol as described at http://www.broadinstitute.org/rnai/public/resources/protocols. Infected cells (30 h post-infection) were selected in puromycin or blasticidin for 2–4 days (2 days for RNA-Seq) and harvested immediately for the experiments.

**Compounds and antibodies.** Palbociclib (S1116), Abemaciclib (S7158), and Ribociclib (S7440) were purchased from Selleck Chemicals (Houston, TX, USA). Antibodies against HSP90 (H-114), Cyclin D1 (M20), CDK6 (C-21), CDK4 (DCS-35), p16 (C-20), p21 (H164), p27 (C19), and cyclin E (HE12) were from Santa Cruz Biotechnology; antibodies against Cyclin D2 (D52F9) and p-RB (S795) were from Cell Signaling; antibody against SMARCA4 were from Bethyl Laboratories (A300-813A). Antibody against RB (554136) was from BD Pharmingen. Cyclin D3 (ab28283) antibody was from Abcam. Antibody against SMARCA4 was used with 1:5000 dilution and all others with 1:1000 dilution. Antibodies for IHC are listed in the corresponding method section below.

**Plasmids.** Individual shRNA vectors used were from the Mission TRC library (Sigma) provided by Genetic Perturbation Service (GPS) of Goodman Cancer Research Centre and Biochemistry at McGill University: shCDK6#1

(TRCN0000010081); shCDK6#2 (TRCN0000009878); shCDK6#3 (TRCN0000010473); shCCND1#1 (TRCN0000295876), shCCND1#2 (TRCN0000288598); and shCDK4#1 (TRCN0000000362), shCDK4#2 (TRCN0000196986), shCDK4#3 (TRCN0000197041). pLX304-GFP, pLX304-CDK6, pLX304-CCND1, pLX317-GFP, and pLX317-CDK4 were obtained from TRC3 ORF collections from TransOMIC and Sigma provided by GPS. pReceiver-Lv120 and pReceiver-Lv120-SMARCA4 were purchased from GeneCopoeia. pIN20 and pIN20-SMARCA4[57] were provided by Dr. Jannik N. Andersen (The University of Texas, MD Anderson Cancer Center).

CDK4[D158N] and CDK6[D163N] kinase-inactive mutant (PMID: 17420273) constructs were generated by site-directed mutagenesis. pLX317-CDK4 and pLX304-CDK6 were used as templates and PCR amplified using Phusion High-Fidelity DNA Polymerase (Thermo Fisher) with following primers designed by NEBaseChanger (NEB).

pLX317-CDK4[D158N]-Forward: CAAGCTGGCTAACTTTGGCCT
pLX317-CDK4[D158N]-Reverse: ACTGTTCCACCACTTGTC
pLX304-CDK6[D163N]-Forward: AAAACTCGCTAACTTCGGCCTTG
pLX304-CDK6[D163N]-Reverse: ATTTGTCCGCTGCTGGTCTC

PCR products were treated with KLD Enzyme Mix (NEB) and transformed into competent cells. Clones were verified by sequencing before experiments.

**Pooled shRNA screen.** Synthetic lethal screens using an shRNA library targeting human kinases and additional kinase-related genes constructed from the TRC human genome-wide shRNA collection (TRC-Hs1.0) were performed as described[28–30]. The screen data were processed using the MAGeCK statistical software package (version 0.5.4)[31].

**Colony-formation assays.** Single-cell suspensions of all cell lines were seeded into 6-well plates (2–8 × 10^4 cells per well depending on proliferation rate and cell size) and cultured both in the absence and presence of drugs as indicated. At the end points of colony-formation assays, cells were fixed with 3.75% formaldehyde, stained with crystal violet (0.1% w/v), and photographed. All relevant assays were performed independently at least three times.

**Cell viability assays.** Cultured cells were seeded into 96-well plates (1000–4000 cells per well). Twenty-four hours after seeding, serial dilutions of palbociclib were added to cells to final drug concentrations ranging from 0.0026–4 μM. Cells were then incubated for 5–10 days and cell viability was measured using the CellTiter-Blue viability assay (Promega). Relative survival in the presence of palbociclib was normalized to the untreated controls after background subtraction.

**Protein lysate preparation and immunoblots.** Cells were first seeded in normal medium without inhibitors. After 24 h, the medium was replaced with fresh medium containing the inhibitors as indicated in the text. After the drug stimulation, cells were washed with cold phosphate-buffered saline (PBS), lysed with protein sample buffer, and processed with Novex® NuPAGE® Gel Electrophoresis Systems (Invitrogen).

**Immunoprecipitation and kinase assay.** Cells were resuspended in ice-cold lysis buffer (50 mM Tris pH 7.5, 150 mM NaCl, 1% NP40, 1 mM dithiothreitol (DTT), and protease/phosphatase inhibitors) and broken by passing through 20-gauge needles 20 times. After 30 min incubation on ice, lysates were clarified by centrifugation at 14,000 × g for 15 min at 4 °C. Supernatant was collected as cell extract and protein concentrations were determined using Bradford Protein Assay (Bio-Rad). Three micrograms of HA (F-7, Santa Cruz) or CDK4 (DCS-35, Santa Cruz) antibodies were added to 2 mg of pre-cleared cell lysate in 500 µl of lysis buffer and incubated overnight at 4 °C with continuous rocking. Protein immunocomplexes were then incubated with 40 µl protein G sepharose beads (Protein G Sepharose 4 Fast Flow, GE Healthcare) at 4 °C for 2 h. Precipitated proteins were washed three times with lysis buffer and eluted with sodium dodecyl sulfate (SDS)-loading buffer at 95 °C for 10 min and analyzed by western blot using Novex® NuPAGE® Gel Electrophoresis Systems (Invitrogen).

For kinase assays, precipitated proteins were washed for another three times with kinase buffer (50 mM HEPES pH 7.5, 10 mM MgCl2, 1 mM DTT, 2 mM CaCl2, 1 mM Imidazole, 1 mM MgAcetate, 1 mM NaF, and 0.5 mM Na3VO4) and then incubated for 40 min at 30 °C in 30 µl kinase buffer containing 5 µCi [32P] ATP (PerkinElmer), 50 µM ATP (Thermo Fisher) and 1 µg recombinant human Rb protein (ab56270, Abcam). Reactions were stopped by placing them on ice followed by addition of 6× SDS-loading buffer. Heat-denatured samples were analyzed on a 10% SDS-polyacrylamide gel electrophoresis, dried, and exposed to Biomax XAR film (Sigma-Aldrich, F5513).

**Cell line RNA-Seq.** Total RNA from cell lines was extracted with the RNeasy Mini Kit (Qiagen, Hilden, Germany) and quality controlled and subjected for RNA-Seq at Genome Quebec and the Institute for Research in Immunology and Cancer at Université de Montréal. Sequencing reads were mapped to reference human genome sequence (hg19) downloaded from Illumina iGenomes using STAR[58] (version 2.4.2). The number of fragments was counted with HTSeq[59] (version 0.6.1p1)

based on known gene from RefSeq database on UCSC Genome Browser. Differential expression of genes was analyzed by Bioconductor package DESeq2[60] (version 1.19.38). Differently expressed genes were visualized in heatmap using R package gplots (version 3.0.1). Gene ontology biological process was performed using the GSEA[61] provided by Broad Institute. Among the enriched gene signatures, the top ten signatures were presented as bar plots according to p value, with GSEA plots for the top three signatures.

**Chromatin immunoprecipitation.** Cells were fixed in complete media with 1% formaldehyde for 10 min and then quenched by addition of 0.125 M glycine for 5 min at room temperature and 15 min on ice. Fixed cells were then pelleted and washed once with 1× PBS before snap-freezing on dry-ice. Cell pellets were lysed in three successive buffers for 10 min each at 4 °C while rotating end-over-end (LB1: 50 mM HEPES-KOH pH 7.5, 120 mM NaCl, 1 mM EDTA, 10% glycerol, 0.5% NP-40, 0.25% Triton X-100, LB2: 10 mM Tris-HCl pH 8.0, 200 mM NaCl, 1 mM EDTA, 0.5 mM EGTA, LB3: 10 mM Tris-HCl pH 8.0, 100 mM NaCl, 1 mM EDTA, 0.5 mM EGTA, 0.1% Na-Deoxycholate, 0.5% N-lauroylsarcosine). Lysates were then sonicated with a Branson450D cup-horn system to produce chromatin fragments between 100 and 600 bps. Triton X-100 was added to cell lysate and then centrifuged at 20,000 × g for 15 min at 4 °C to pellet debris. Supernatant equivalent to 10 million cells brought to a final volume of 500 µL using LB3 was used for each immunoprecipitation. A quantity of sonicated chromatin was set aside as 10% input. Two micrograms of antibody was added to the lysate for overnight incubation at 4 °C. Antibodies used were immunoglobulin G (IgG; abcam ab37415) and αBRG1 (Bethyl A300-813A). Protein G Magnetic Dynabeads® (ThermoFisher Scientific) were used for pull down. Immunoprecipitated chromatin bound to dynabeads was washed with four successive buffers (LSB: 20 mM Tris-HCl pH 8.0, 150 mM NaCl, 0.1% SDS, 1% Triton X-100, 2 mM EDTA; MSB: 20 mM Tris-HCl pH 8.0, 250 mM NaCl, 0.1% SDS, 1% Triton X-100, 2 mM EDTA; LiCl wash: 10 mM Tris-HCl pH 8.0, 250 mM LiCl, 0.5% NP-40, 0.5% Na-deoxycholate, 1 mM EDTA; 1× TE: 10 mM Tris-HCl pH 8.0, 1 mM EDTA). Chromatin was then eluted from dynabeads in 150 µL EB (50 mM Tris-HCl pH 8.0, 10 mM EDTA, 1% SDS) by incubating at 65 °C for 30 min. Immunoprecipitated samples and input were incubated overnight at 65 °C to denature formaldehyde crosslinking. Samples were then treated with RNaseA (ThermoFisher Scientific) followed by proteinase K (Sigma Aldrich) before phenol/chloroform extraction. DNA was precipitated using 5 M NaCl, glycoblue (Ambion), and 100% absolute ethanol overnight at −80 °C. DNA was pelleted by centrifuging at 20,000 × g for 30 min at 4 °C followed by a 70% ethanol wash. Final DNA pellet was resuspended in 50 µL of 1× TE buffer and placed in speedvac for 3 min.

**RNA isolation and quantitative reverse transcriptase-PCR (qRT-PCR).** For cell line samples, cells were first seeded and then harvested for RNA isolation using Trizol (Invitrogen) the next day.

Synthesis of cDNAs and qRT-PCR assays were carried out to measure the mRNA levels of genes as described[29]. Relative mRNA levels of each gene shown were normalized to the expression of the house-keeping gene GAPDH. The sequences of the primers for assays using SYBR® Green master mix (Roche) are as follows:

GAPDH_Forward, AAGGTGAAGGTCGGAGTCAA;
GAPDH_Reverse, AATGAAGGGGTCATTGATGG;
CCND1_Forward, GGCGGATTGGAAATGAACTT;
CCND1_Reverse, TCCTCTCCAAAATGCCAGAG;
CCND2_Forward, ACGGTACTGCTGCAGGCTAT;
CCND2_Reverse, AGCTGCTGGCTAAGATCACC;
CCND3_Forward, TTGAGCTTCCCTAGGACCAG;
CCND3_Reverse, TGACCATCGAAAAACTGTGC;
CDK4_Forward, GTCGGCTTCAGAGTTTCCAC;
CDK4_Reverse, TGCAGTCCACATATGCAACA;
CDKN2A Exon 1a_Forward, CATAGATGCCGCGGAAGGT;
CDKN2A Exon 1a_Reverse, CCCGAGGTTTCTCAGAGCCT;
CCNE1_Forward, TCTTTGTCAGGTGTGGGGA;
CCNE1_Reverse, GAAATGGCCAAAATCGACAG;
CDKN1B_Forward, AACGTGCGAGTGTCTAACGG;
CDKN1B_Reverse, CCCTCTAGGGGTTTGTGATTCT;
CDKN1A_Forward, CCTGTCACTGTCTTGTACCCT;
CDKN1A_Reverse, GCGTTTGGAGTGGTAGAAATCT;
ESR1_Forward, CAGGATCTCTAGCCAGGCAC;
ESR1_Reverse, ATGATCAACTGGGCGAAGAG;

TaqMan® Gene Expression Assays probes CDK4 (Hs00364847_m1) and CDK6 (Hs01026371_m1) were purchased from Thermo Fisher Scientific.

The primers used for the chromatin immunoprecipitation were designed guided by publicly available Encode ChIP-seq tracks of SMARCA4 and are as follows:
CCND1 Promoter Fwd, CCGGAATGAAACCTTGCACAGG;
CCND1 Promoter Rev, AGACGGCCAAAGAATCTCAGC;
CCND1 Upstream 3 Fwd, AAGTCACTCTTCCGTAGAGC;
CCND1 Upstream 3 Rev, GGCACCTGGACCTTCAACAC;
CCND1 Upstream 2 Fwd, GACACAGTAAACAGCACCAG;
CCND1 Upstream 2 Rev, GACTTGTGCCTGTTACACC;
CCND1 Upstream 1 Fwd, ACTGGAGAGAGAGACTGATTGC;

*CCND1* Upstream 1 Rev, CCAGGCCAGTCTCATTACTG;
*CCND3* Promoter Fwd, CCTCCCATTTTGCTTCTCGG;
*CCND3* Promoter Rev, TGAGTCATTACATCGTGAGG;
*CCNE1* Promoter Fwd, GGCTTTAAGTGAGAGATGGG;
*CCNE1* Promoter Rev, TTCATCCGTCAGTGCATTGG;
GAPDH_Forward, AAGGTGAAGGTCGGAGTCAA;
GAPDH_Reverse, AATGAAGGGGTCATTGATGG.

**Mouse xenograft and PDX drug studies**. For in vivo drug studies, palbociclib (SelleckChem, S1116) was resuspended in 50 mM sodium L-lactate (Sigma Aldrich) buffer (pH = 4.0), at a concentration of 15.75 mg mL$^{-1}$ (150 mg kg$^{-1}$ dose for a 21 g mouse in a volume of 200 µL) and stored at $-80$ °C. Tubes were thawed overnight at 4 °C. Animal experiments were carried out according to standards outlined in the Canadian Council on Animal Care Standards (CCAC) and the Animals for Research Act, R.S.O. 1990, Chapter c. A.22 and by following internationally recognized guidelines on animal welfare.

All BIN-67 animal experiments were carried out at the University of Ottawa using 7–8-week-old female CB-17 SCID mice (CB17/lcr-Prkdc$^{scid}$/lcrlcoCrl strain code 236, Charles River). BIN-67 cells, $10 \times 10^6$ in 100 µL 50:50 PBS:Matrigel (Cultrex 3D Culture Matrix™ BME, Trevigen Inc) were subcutaneously injected into the shaved right flank of each of 16 SCID mice, average weight 20.7 g (SE = 0.16 g). When tumor volumes (TVs) ($V = (H \times W^2)/2$) reached ~120 mm$^3$ (4 weeks post inoculation), which was assigned as Day 1, the mice were entered into the treatment regimen (200 µL p.o.×21 days). Eight mice were each randomly allocated to vehicle control (50 mM sodium L-lactate buffer, pH 4.0) or treatment group (150 mg kg$^{-1}$ palbociclib).

All SCCOHT-1 animal experiments were carried out at the Research Institute of McGill University Health Centre, using 8–12-week-old female NOD-scid/IL2Rg$^{null}$ mice (NOD.Cg-Prkdc$^{scid}$Il2rg$^{tm1}$/SzJ, 005557, Jackson Laboratories, USA). For the SCCOHT-1 tumor model, a single-cell suspension was created by dissociating a sufficient number of sub-confluent flasks of cells to produce 6 million cells in 200 µL of Matrigel HC in a 50:50 ratio (Corning Matrigel HC, Cat #. 354428, VWR, Mississauga, Canada). The tumor cell suspension was subcutaneously injected into the left flank of each NSG mouse, average weight 24.1 g (SE = 2.4). When TVs ($V = (H \times W^2)/2$) reached ~60 mm$^3$ (3 weeks post inoculation), which was assigned as Day 1, the mice were entered into the treatment regimen (200 µL p.o. × 42 days). Four mice were each randomly allocated to vehicle control (50 mM sodium L-lactate buffer, pH 4.0) or 5 mice to the treatment group (150 mg kg$^{-1}$ palbociclib).

For both BIN-67 and SCCOHT-1 models, mice were housed in groups of 3–5, with each group consisting of both vehicle control and treatment animals matched for tumor size on Day 1 of treatment. All gavage treatments were carried out using sterile straight 20- or 22-gauge, 38.1 mm stainless steel feeding tubes (Harvard Apparatus, QC). Tumor progression was monitored and measurements using digital calipers (VWR International) were recorded twice weekly. The persons who performed all the tumor measurements and the IHC analysis for the end point tumor samples were blinded to the treatment information.

The SCCOHT PDX was established and viably preserved at Memorial Sloan Kettering Cancer Center (MSKCC). The PDX establishment was performed in accordance with the Institutional Animal Care and Use Committee at MSKCC. The patient tumor sample was received from the University of North Carolina with the Institutional review board approval and informed consent from the patient. The PDX work described in this manuscript was undertaken at NYU Langone Health and use of animals was overseen by the Division of Comparative Medicine under the direction and approval of the Institutional Animal Care and Use Committee at that institution and was conducted in accordance with all pertinent Federal regulations and policies. The experimental protocol used in these studies was approved by the Institutional Animal Care and Use Committee at NYU Langone Health. We used 5–6-week-old female mice: strain NOD.Cg-Prkdc$^{scid}$IL2rg$^{tm1Wjl}$/SzJ (Jackson Laboratory; ref. #005557).

The viably frozen PDX was thawed and initially implanted in six mice for each treatment arm in the animal facility at NYU Langone Health as follows. The frozen PDX was thawed in PBS and washed twice to remove residual dimethyl sulfoxide. Tumor cells (300 µL) were then injected intraperitoneally using a 16-gauge needle. Once the tumors developed (approximately 3 months after injection), they were further propagated in multiple mice by subcutaneous injection. Single-cell tumor suspension was mixed with matrigel in a 1:1 ratio (50:50 µL) and injected in the flanks. Once tumors reached approximately 100 mm$^3$ in volume, we randomized animals into two groups; vehicle-treated and palbociclib-treated. As a vehicle, we used 50 mM sodium L-lactate (pH4). Palbociclib was solubilized in vehicle to a final concentration 15 mg mL$^{-1}$ and stored at $-80$ºC. Treatments were performed daily by oral gavage with 200 µL (3 mg) of palbociclib solution. Mice were treated with the initial dose of 150 mg kg$^{-1}$ of palbociclib. After first signs of any animal discomfort, we reduced the dose to 100 mg kg$^{-1}$. Mice that showed weight loss were not included for the analysis. TVs and weight were measured twice a week. To calculate TVs, we used caliper to measure the height ($H$), length ($L$), and width ($W$) and followed this formula: TV = ($H \times L \times W$)/2.

**Patient tumor samples**. Tumor samples of 54 different SCCOHT patients were used in this study (Fig. 6). The *SMARCA4* mutation status of 51 cases were

confirmed by DNA analysis. Three cases that do not have DNA mutation information were confirmed for SMARCA4 protein loss by IHC.

Studies on SCCOHT patient tumors ($n = 33$) were approved by the Institutional Review Board (IRB) at McGill University, McGill IRB # A08-M61-09B. Of these 33 SCCOHTs, 5 cases had fresh frozen samples for qPCR analysis (Fig. 6c) and 32 cases were used in making the TMA (Fig. 6d, e). Studies on 59 pathologist-confirmed (B.A.C.) ovarian HGSC samples ($n = 7$, Montreal, Fig. 6c and $n = 52$, Toronto, Fig. 6d, e) were approved by the ethics boards at the University Hospitals Network and the Jewish General Hospital respectively. Informed consent was obtained from all participants in accordance with the relevant IRB approvals. Hematoxylin and eosin (H&E)-stained sections of the 32 SCCOHTs (confirmed by DNA mutation analysis or/and SMARCA4 IHC) and 52 HGSC cases were reviewed and areas containing tumor only were demarcated and cored to construct TMAs using duplicate 0.6-mm cores from the demarcated areas.

The NanoString gene expression studies using 17 SCCOHT[8,50] and 6 HGSC patient samples were approved by the IRB at MSKCC (Fig. 6a, b). Specialty gynecologic pathologists reviewed the cases to confirm diagnosis using previously described guidelines[8]. RNA was extracted from formalin-fixed paraffin-embedded (FFPE) tumor sections with at least 50% tumor cell content. For the RNA extraction, we used the Ambion's RecoverAll™ Total Nucleic Acid Isolation Kit for FFPE (Cat# AM1975) and performed extraction according to the manufacturer's suggestions. Subsequently, the RNA quality was confirmed using the Agilent's Bioanalyzer and the RNA amounts applied to the platform were adjusted according to the Quality Control score.

**Gene expression analysis**. For the NanoString gene expression (nCounter Pan-Cancer Pathways Panel) study, the raw reporter code count data were normalized using the NanoStringNorm (version 1.1.21) R package. VSN affine transformation was applied to stabilize variance to normalize the data for visualization. Heatmap visualization was with ComplexHeatmap[62] (version 1.10.2).

**Immunohistochemistry**. For mouse xenografts, 4-micron-thick sections from FFPE tissue were cut, deparaffinized, and stained using an IntelliPath automated immunostainer (Biocare Medical). The protocol included an antigen retrieval treatment in Diva Decloaker RTU (Biocare Medical) for 10 min followed by incubation with the primary antibody (phosphoRB, Cell Signaling, 9308, 1/200 dilution; KI-67, Abcam, 16667, 1/100 dilution) for 1 h at room temperature. Incubation was followed by detection using a Goat anti Rabbit horseradish peroxidase (Dako) and 3,3'diaminobenzidine (DAB; Dako). The slides were digitalized using an Aperio scanner. The mitotic index was measured by counting the mitotic active cells in 10 high-power fields (×400) of the H&E-stained tumor slides.

Patient TMAs were stained with the following primary IHC antibodies Cyclin D1 (rabbit polyclonal, AbCam, 1:100), CDK4 (rabbit polyclonal, AbCam, 1:100), CDK6 (rabbit polyclonal, AbCam, 1:500), RB (mouse monoclonal, Cell Signaling 9309, 1:300), and p16 (mouse monoclonal, MTM Laboratories, prediluted). Sections were treated with xylene (EMD Chemicals Gibbstown, NJ) for 30 min and rinsed in xylene for 5 min each and then rehydrated in a series of descending concentrations of ethanol (Fisher Scientific Fair Lawn, NJ). Slides were then treated with 0.3% H$_2$O$_2$/methanol (AppliChem Darmstadt, Germany/Fisher Scientific Fair Lawn, NJ) for 30 min to block endogenous peroxide. Heat-induced epitope retrieval was achieved by treating slides in a pressure cooker with 0.01 M citrate buffer (Vector Burlingame, CA) (pH 7.6). Slides were then rinsed in 0.1 M Tris buffer (Dako Caprinteria, CA) with Tween 20 (Dako Caprinteria, CA), then blocked with 2% FBS (Sigma-Aldrich St. Louis, MO) for 5 min. Slides were subsequently incubated with the specified primary antibodies for 1 h at room temperature or overnight at 4 °C. Slides were then rinsed in Tris (Dako Caprinteria, CA) and then incubated with biotinylated anti-mouse IgG or anti-goat IgG (Vector Laboratories; U0625) for 30 min at room temperature. After rinsing in Tris, slides were incubated with the avidin–biotin complex (Vector Laboratories; PK-6100) for 30 min at room temperature, followed by incubation with DAB (DAKO Cytomation; K3468) for 10 min at room temperature. Slides were then rinsed, dehydrated through a series of ascending concentrations of ethanol and xylene, and coverslipped.

**Automated quantification**. Both SCCOHT and HGSC TMAs were scanned using an Aperio Scanscope Scanner (Aperio Vista, CA) and viewed through the Aperio ImageScope software program. An individual blinded to the experimental design captured JPEG images from each core (circular area of 315 cm$^2$ corresponding to the entire core) at ×10 magnification on the Aperio ImageScope viewing program. The same blinded individual performed quantification of immunostaining on each JPEG using an automated analysis program with Matlab's image processing toolbox based on previously described methodology[52]. Cores with low tumor cellularity and artifacts were not included in the analysis. The algorithm used color segmentation with RGB color differentiation, K-Means Clustering, and background–foreground separation with Otsu's thresholding. To arrive at a score for each core (represented as pixel units), the cases were unblinded and the number of extracted pixels were multiplied by their average intensity. The final score for a given case and marker was calculated by averaging the score of two cores (for each case) for a given marker.

**Statistical analysis**. Statistical significance was calculated by two-tailed Student's $t$ test, two-way analysis of variance, or non-parametric Mann–Whitney test accordingly. Prism 6 software was used to generate graphs and statistical analyses. $*p < 0.05$, $**p < 0.01$, $***p < 0.001$, $****p < 0.0001$.

## Data availability

The RNA-Seq data have been deposited to the Gene Expression Omnibus (GEO) with accession number GSE120297 and GSE120298. Uncropped western blots for the most important experiments are displayed in Supplementary Fig. 12. All other data supporting the findings of this study are available by contacting the corresponding authors on reasonable request.

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

## Acknowledgements

We thank K.A. Orlando, J.R. Raab, and B. Weissman for sharing ChIP-seq data supporting our findings. We thank O. Collins for technical assistance, S. Albrecht for pathology discussions, and C. Pochet and C. Sun for support. This work was supported by the Canadian Institutes of Health Research (CIHR) grants MOP-130540 (to S.H.) and PJT-156233 (to S.H.); a Merck, Sharpe & Dohme Corp./McGill Faculty of Medicine Grant for Translational Research (to S.H. and W.D.F.); a Canadian Cancer Research Institute Innovation grant (#702961, to W.D.F.); a USA Department of Defense Office of Congressionally Directed Medical Research Grant (#W81XWH-15-1-0497, to W.D.F. and S.H.); a CIHR Foundation Grant (FDN-148390) (to W.D.F); a CIHR Foundation Grant (FDN-148366) (to J.P.) and a CIHR Foundation Grant (FDN-143322) (to J.R.); and grants from the Cancer Research Society (to B.V. and W.D.F), the Small Cell Ovarian Cancer Foundation (to B.V. and W.D.F), and an educational grant from Fondation Gustave Roussy (to A.L.). Y.X. and A.I.P. were supported by CIHR-funded Chemical Biology Scholarships. Y.X. is supported by a Maysie MacSporran Graduate Studentship and a Cedars Cancer Institute Fellowship and S.H. is supported by a CRC Chair in Functional Genomics.

## Author contributions

Y.X., B.M., E.M., S.V., X.Q.D.W., L.W., T.K., D.M., G.M., R.C., A.A., H.C., A.I.P. and N.B. performed experiments. Y.X., E.M., A.O., I.d.R and R.M.J performed statistical analyses. P.J., A.A., W.H.G., B.A.C., A.L. and D.A.L. contributed samples and technical support. S.V., D.M., B.A.C., M.-C.G., and A.R.J. provided pathology expertise. R.M.K. A.L., J.P., J.D., M.P, D.A.L., A.R.J., R.H., J.R. and B.V. oversaw the experiments and provided advice. Y.X., W.D.F. and S.H. designed the experiments, analyzed the data, and wrote the manuscript. W.D.F. and S.H. conceived and oversaw the study. All authors read and approved the final manuscript.

## Additional information

**Competing interests:** The authors declare no competing interests.

Yibo Xue [1,2], Brian Meehan[3,4], Elizabeth Macdonald[5,6], Sriram Venneti[7], Xue Qing D. Wang[1], Leora Witkowski[8,9,10,11], Petar Jelinic [12], Tim Kong[1,2], Daniel Martinez[13], Geneviève Morin[1,2], Michelle Firlit[12], Atefeh Abedini[5,6], Radia M. Johnson[1,2], Regina Cencic[1,2], Jay Patibandla[12], Hongbo Chen[14], Andreas I. Papadakis[1,2], Aurelie Auguste[15], Iris de Rink[16], Ron M. Kerkhoven[16], Nicholas Bertos[1,2], Walter H. Gotlieb[17], Blaise A. Clarke[18], Alexandra Leary[15], Michael Witcher[19,20,21,22], Marie-Christine Guiot[23], Jerry Pelletier[1,2], Josée Dostie[1], Morag Park[1,2], Alexander R. Judkins[24], Ralf Hass [25], Douglas A. Levine [12], Janusz Rak[3,4], Barbara Vanderhyden[5,6], William D. Foulkes [8,9,10,11] & Sidong Huang [1,2]

[1]Department of Biochemistry, McGill University, Montreal, QC H3G 1Y6, Canada. [2]The Rosalind & Morris Goodman Cancer Research Centre, McGill University, Montreal, QC H3A 1A3, Canada. [3]Department of Pediatrics, McGill University, Montreal, QC H4A 3J1, Canada. [4]Research Institute of McGill University Health Centre Montreal Children's Hospital, Montreal, QC H4A 3J1, Canada. [5]Centre for Cancer Therapeutics, Ottawa Hospital Research Institute, Ottawa, ON K1Y 4E9, Canada. [6]Department of Cellular and Molecular Medicine, University of Ottawa, Ottawa, ON K1H 8M5, Canada. [7]Pathology and Neuropathology, University of Michigan Medical School, Ann Arbor, MI 48109-0605, USA. [8]Department of Human Genetics, McGill University, Montreal, QC H3A 0C7, Canada. [9]Department of Medical Genetics, Jewish General Hospital, McGill University, Montreal, QC H3T 1E2, Canada. [10]Lady Davis Institute, McGill University, Montreal, QC H3T 1E2, Canada. [11]Department of Medical Genetics and Cancer Research Program, Research Institute of the McGill University Health Centre, McGill University, Montreal, QC H4A 3JI, Canada. [12]Gynecologic Oncology, Laura and Isaac Perlmutter Cancer Center, NYU Langone Medical Center, New York, NY 10016, USA. [13]Children's Hospital of Philadelphia Research Institute, Philadelphia, PA 19104, USA. [14]School of Pharmaceutical Sciences (Shenzhen), Sun Yat-Sat University, 510275 Guangzhou, China. [15]Department of Cancer Medicine, Gustave Roussy, INSERM U981, 94800 Villejuif, France. [16]Genomics Core Facility, The

Netherlands Cancer Institute, 1066 CX Amsterdam, The Netherlands. [17]Division of Gynecologic Oncology, Segal Cancer Center, Jewish General Hospital, McGill University, Montreal, QC H3T 1E2, Canada. [18]Department of Laboratory Medicine and Pathobiology, University of Toronto, University Health Network, Toronto, ON M5G 2C4, Canada. [19]Department of Oncology, McGill University, Montreal, QC H3T 1E2, Canada. [20]Department of Experimental Medicine, McGill University, Montreal, QC H3T 1E2, Canada. [21]Lady Davis Institute, Jewish General Hospital, Montreal, QC H3T 1E2, Canada. [22]Segal Cancer Centre, Jewish General Hospital, Montreal, QC H3T 1E2, Canada. [23]Department of Pathology, Montreal Neurological Hospital/Institute, McGill University Health Centre, Montreal, QC H3A 2B4, Canada. [24]Department of Pathology and Laboratory Medicine, Children's Hospital Los Angeles, Keck School of Medicine of University of Southern California, Los Angeles, CA 90027, USA. [25]Biochemistry and Tumor Biology Laboratory, Department of Gynecology and Obstetrics, Medical University Hannover, 30625 Hannover, Germany

