## [Peer Review File · Nature Communications]

Reviewers' comments:

Reviewer #1 (Remarks to the Author):

This is a revised manuscript transferred from Nature Genetics. The authors have performed a few additional experiments to address the reviewers' points following the Editors suggestions. While overall the manuscript is approved, I find the authors explanation for why not include SMARCA4 ChIP-seq data not satisfactory. The reason to do ChIP-seq is not just to show that other genes are also targeted by SMARCA4, but to provide more conclusive proof that the authors ChIP-PCR data is actually real and the antibody/ChIP-seq passes necessary quality controls. Many times one can get an antibody stick to a particular region more than to another and still be just an artifact of antibody/protocols. Even the fact that despite multiple requests the authors do not include this experimental data raises concerns, but if the Editors deem that such data validation is not necessary then I have no further comments on this.

Reviewer #3 (Remarks to the Author):

The authors conclude that SCCOHT tumors, driven by inactivating SMARCA4 mutations, also demonstrate diminished Cyclin D1 levels. Cyclin D1 loss reduces the kinase activity of Cdk4 and conveys sensitivity to Cdk4/6 inhibitors. Overall, the authors have adequately addressed all prior concerns of this reviewer. Since the initial version, significant improvements to the manuscript include incorporation of new mechanistic details that support the conclusions and more clearly elucidate the relationship between SMARCA4 loss and Cdk4/6 inhibitor sensitivity in SCCOHT.

Minor concerns:

1. The main conclusion of the manuscript, that diminished but residual cyclinD1 levels resulting from SMARCA4-deficiency convey particular sensitivity to Cdk4/6 inhibitors, is now strongly supported within a suite of expanded experiments. However, this conclusion is a bit difficult to understand in the Results and Discussion sections. I recommend focusing the discussion of this conclusion to emphasize this critical finding and make direct references to the critical pieces of data supporting it. Convincing arguments were made in author responses, and should be utilized for incorporation into the manuscript.
2. Similarly, a convincing discussion of why cyclin D2 was deemed non-essential for Cdk4/6 inhibitor response would be helpful. This was also described in more detail in the author response. Is any data available from NanoString data or IHC against cyclin D2 to further support the conclusions?
3. On page 8, line 16, in the phrase "retain functional RB", it is not clear whether this is known from prior literature or characterization of these lines or whether this data is demonstrated by the Western blot referenced (Fig. 3A). If the former, clarification and/or citation would be useful, if the latter, the blot does not convincingly demonstrate function of RB.
4. Most mentions of "correlation" of gene or protein expression (e.g. on page 8, lines 19-21) do not incorporate any correlative statistics or measurement. I suggest that this language is changed

to reflect the association of representative cell lines with certain phenotypes, rather than the suggestion of correlative analyses having been performed.

5. On page 9, line 22, the authors describe elevated RB phosphorylation demonstrated in the IP data shown in Figs. 3d and 3e, but these bands are not evident in the immunoprecipitated sample to demonstrate direct association with the Cdk4 complex. Either a longer exposure is needed or reference to the Western blots in Figs. 3f and 3g should be made instead of an implication of association with the complex.

Referee #1: Cancer genomics

Referee #2: Cancer genomics, cell cycle kinases

Reviewers' Comments:

Reviewer #1:

Remarks to the Author:

This is a revised manuscript transferred from Nature Genetics. The authors have performed a few additional experiments to address the reviewers' points following the Editors suggestions. While overall the manuscript is approved, I find the authors explanation for why not include SMARCA4 ChIP-seq data not satisfactory. The reason to do ChIP-seq is not just to show that other genes are also targeted by SMARCA4, but to provide more conclusive proof that the authors ChIP-PCR data is actually real and the antibody/ChIP-seq passes necessary quality controls. Many times one can get an antibody stick to a particular region more than to another and still be just an artifact of antibody/protocols. Even the fact that despite multiple requests the authors do not include this experimental data raises concerns, but if the Editors deem that such data validation is not necessary then I have no further comments on this.

We agree with the reviewer that it is important to rule out the antibody-specific effect.

We have included a SMARCA4 ChIP-seq data set of BIN-67 SCCOHT cells before and after SMARCA4 restoration. This data set was independently generated by the group of Dr. Bernard Weissman using an anti-SMARCA4 antibody (Abcam), which is different than the one that we used (Bethyl). As shown in **Fig. R1a** (attached to this document), SMARCA4 occupancy at *CCND1* promoter was observed in BIN-67 cells upon SMARCA4 restoration, thus independently validating our previous findings.

We include these results as “reviewer only”, because Dr. Weissman generated this data set for a separate manuscript of his group currently under preparation. This study was recently presented at the 2018 AACR annual meeting:

http://cancerres.aacrjournals.org/content/78/13_Supplement/4318

In addition, we show that this regulation of *CCND1* by SMARCA4 is conserved in lung cancer in the accompanying manuscript. Supporting our findings, a ChIP-seq data set of SMARCA4-deficient H1299 lung cancer cells engineered to express inducible exogenous SMARCA4 has been recently published (**Fig. R1b**). SMARCA4 occupancy at *CCND1* promoter was also observed upon SMARCA4 induction.

Furthermore, we also attached a summary of additional publicly available SMARCA4 ChIP-Seq data sets of 6 cell lines of different tissue origins, generated by 4 other groups (**Fig. R2**). Overall, these data show consistent SMARCA4 occupancy at the *CCND1* promoter region and thus further support our findings.

Reviewer #3:

Remarks to the Author:

The authors conclude that SCCOHT tumors, driven by inactivating SMARCA4 mutations, also demonstrate diminished Cyclin D1 levels. Cyclin D1 loss reduces the kinase activity of Cdk4 and conveys sensitivity to Cdk4/6 inhibitors. Overall, the authors have adequately addressed all prior concerns of this reviewer. Since the initial version, significant improvements to the manuscript include incorporation of new mechanistic details that support the conclusions and more clearly elucidate the relationship between SMARCA4 loss and Cdk4/6 inhibitor sensitivity in SCCOHT.

We appreciate that the reviewer recognizes and approves the significant improvements of our manuscript.

Minor concerns:

1. The main conclusion of the manuscript, that diminished but residual cyclinD1 levels resulting from SMARCA4-deficiency convey particular sensitivity to Cdk4/6 inhibitors, is now strongly supported within a suite of expanded experiments. However, this conclusion is a bit difficult to understand in the Results and Discussion sections. I recommend focusing the discussion of this conclusion to emphasize this critical finding and make direct references to the critical pieces of data supporting it. Convincing arguments were made in author responses, and should be utilized for incorporation into the manuscript.

We thank the reviewer for this suggestion and have improved the results and discussion sections accordingly. For example, please see the second and third paragraphs of the discussion section.

2. Similarly, a convincing discussion of why cyclin D2 was deemed non-essential for Cdk4/6 inhibitor response would be helpful. This was also described in more detail in the author response. Is any data available from NanoString data or IHC against cyclin D2 to further support the conclusions?

As suggested by the reviewer, we have analyzed *CCND2* expression in the NanoString data set (**new Supplementary Fig. 10a**) and performed additional qRT-PCR analysis of the patient tumor samples (**new Supplementary Fig. 10b, c**). SCCOHT tumors expressed variable levels of *CCND2*, in contrast to HGSCs, which all exhibited low *CCND2* expression. This is consistent with our cell line data showing that SCCOHT-1 but not the other 2 SCCOHT cell lines (or any other cells studied) expresses elevated cyclin D2 (Supplementary Fig. 3a, g), indicating a potential compensatory response to cyclin D1 deficiency.

However, our *in vitro* kinase assays indicate that SCCOHT-1 cells, despite of elevated cyclin D2 expression, still have lower total CDK4 kinase activity compared to SMARCA4-proficient cells (Fig. 3c). Thus, these data suggest that cyclin D2 cannot fully compensate for cyclin D1 deficiency. Alternative compensatory mechanisms for cell cycle progression may play a role in SCCOHT which requires further study. We have incorporated this in the discussion as suggested.

3. On page 8, line 16, in the phrase "retain functional RB", it is not clear whether this is known from prior literature or characterization of these lines or whether this data is demonstrated by the Western blot referenced (Fig. 3A). If the former, clarification and/or citation would be useful, if the latter, the blot does not convincingly demonstrate function of RB.

We have removed “functional” from this phrase to be accurate.

4. Most mentions of "correlation" of gene or protein expression (e.g. on page 8, lines 19-21) do not incorporate any correlative statistics or measurement. I suggest that this language is changed to reflect the association of representative cell lines with certain phenotypes, rather than the suggestion of correlative analyses having been performed.

We agree with the reviewer and have replaced “correlate” with “associate”.

5. On page 9, line 22, the authors describe elevated RB phosphorylation demonstrated in the IP data shown in Figs. 3d and 3e, but these bands are not evident in the immunoprecipitated sample to demonstrate direct association with the Cdk4 complex. Either a longer exposure is needed or reference to the Western blots in Figs. 3f and 3g should be made instead of an implication of association with the complex.

We appreciate that the reviewer points out this unclear description. What we meant is that ectopic cyclin D1 expression elevates RB phosphorylation of the input samples, which suggests increased CDK4/6 kinase activity in these cells. We have modified the text and reference accordingly to clarify this.

ChIP-Seq data sets
using anti-SMARCA4 (Abcam, EPNCIR111A)

a, BIN-67 cells (SCCOHT), +/- SMARCA4 restoration

Orlando et al, manuscript in preparation; Abstract 4318, AACR 2018

Merged tracks of 4 biological replicates (normalized to input).

b, H1299 lung cancer cells (SMARCA4-deficient), inducible SMARCA4

Lissanu Deribe et al, Nature Medicine 2018; PMID: 29892061

Additional publicly available CHIP-Seq data sets using anti-SMARCA4 (Abcam, EPNCIR111A)

- 1) J-Lat A72 cell line (1) GSM267680
- 2) HS-SY-II synovial sarcoma cell line (2) GSM2916100
- 3) 501-Mel melanoma cell line (3) GSM1517753
- 4-6) brain, kidney and liver rhabdoid tumor cell lines GSE71504
Nat Genet 2017 Feb;49(2):289-295. PMID: 27941797

REVIEWERS' COMMENTS:

Reviewer #1 (Remarks to the Author):

The authors have responded to each of the reviewers' specific comments and have revised the manuscript accordingly. The requested CHIP-seq experiments were performed, but not included as this is part of another manuscript. However, since there is public SMARCA4 CHIPseq data in many cell types (CISTROME database has >50 datasets), they could at least show /mention this in the revised paper.

REVIEWERS' COMMENTS:

Reviewer #1 (Remarks to the Author):

The authors have responded to each of the reviewers' specific comments and have revised the manuscript accordingly. The requested ChIP-seq experiments were performed, but not included as this is part of another manuscript. However, since there is public SMARCA4 ChIPseq data in many cell types (CISTROME database has >50 datasets), they could at least show /mention this in the revised paper.

We thank the reviewer for this suggestion using this helpful database. Out of over 50 SMARCA4 ChIP-Seq data sets available through Cistrome, there are 10 different data sets from 8 human cell lines that have passed all quality controls as defined by Cistrome. These ChIP-Seq tracks also show consistent SMARCA4 occupancy at the *CCND1* promoter and thus support our findings. We have included these data in **new Supplemental Figure 9** and have cited Cistrome and these ChIP-Seq studies.